# Multi-omic analysis in transgenic mice implicates omega-6/omega-3 fatty acid imbalance as a risk factor for chronic disease

Kanakaraju Kaliannan[1], Xiang-Yong Li[1], Bin Wang[1], Qian Pan[1], Chih-Yu Chen[1], Lei Hao[1], Shanfu Xie[1] & Jing X. Kang[1]

An unbalanced increase in dietary omega-6 (n-6) polyunsaturated fatty acids (PUFA) and decrease in omega-3 (n-3) PUFA in the Western diet coincides with the global rise in chronic diseases. Whether n-6 and n-3 PUFA oppositely contribute to the development of chronic disease remains controversial. By using transgenic mice capable of synthesizing PUFA to eliminate confounding factors of diet, we show here that alteration of the tissue n-6/n-3 PUFA ratio leads to correlated changes in the gut microbiome and fecal and serum metabolites. Transgenic mice able to overproduce n-6 PUFA and achieve a high tissue n-6/n-3 PUFA ratio exhibit an increased risk for metabolic diseases and cancer, whereas mice able to convert n-6 to n-3 PUFA, and that have a lower n-6/n-3 ratio, show healthy phenotypes. Our study demonstrates that n-6 PUFA may be harmful in excess and suggests the importance of a low tissue n-6/n-3 ratio in reducing the risk for chronic diseases.

[1] Laboratory for Lipid Medicine and Technology, Department of Medicine, Massachusetts General Hospital and Harvard Medical School, Boston, MA 02129, USA. Correspondence and requests for materials should be addressed to J.X.K. (email: kang.jing@mgh.harvard.edu)

Chronic illnesses, including obesity, type 2 diabetes, cardiovascular disease, cancer, and Alzheimer's disease, are rising exponentially in the modern world[1]. These diseases are multi-factorial in nature, but their prevalence coincides with the unbalanced increase in dietary omega-6 (n-6) polyunsaturated fatty acids (PUFA) and decrease in omega-3 (n-3) PUFA in today's diets, suggesting that there may be differential effects of n-6 and n-3 PUFA on the development of chronic disease[2,3]. Recent research has focused mainly on overall PUFA levels or on the level of n-3 PUFA alone, ignoring the important interplay between n-6 and n-3 PUFA levels[4,5]. It is challenging to clarify the differential effects of n-6 and n-3 PUFA or varying n-6/n-3 PUFA ratios due to the confounding factors of diet[6,7]. As a result, studies on PUFA show inconsistent results about the role of PUFA in health and disease[8,9]. Therefore, it is critical to establish a model that allows us to accurately study the importance of the n-6/n-3 ratio and the potential dangers of excess n-6 PUFA[6,10,11].

Chronic low-grade inflammation, often caused by metabolic endotoxemia, is considered to be a critical contributor to the development of many modern chronic diseases[12–16]. Metabolic endotoxemia can often result from gut microbiota dysbiosis and intestinal barrier dysfunction[16]. Previous work has shown that diet is an important modulating factor for both metabolic endotoxemia and chronic low-grade inflammation[16]. Thus, identifying dietary components that can optimize the gut microbiome is important for research into chronic disease prevention and treatment.

The transgenic FAT-1 mouse model was previously developed in our lab to understand the importance of the PUFA ratio in health and disease[6,10]. This mouse model contains the FAT-1 gene from Caenorhabditis elegans, encoding an enzyme that can endogenously convert n-6 to n-3 PUFA[10]. This conversion enables FAT-1 mice to markedly increase their tissue levels of n-3 PUFA and decrease the levels of n-6 PUFA correspondingly, resulting in a significant reduction of tissue ratio of n-6/n-3 PUFA, close to 1:1, with no need for additional supplementation of PUFA in the diet. This unique feature allows for studies to be performed without the confounding factors of diet[6,17], and these mice have been widely used over the last decade to study the beneficial effects of increased tissue levels of n-3 PUFA and balanced (reduced or low) tissue ratio of n-6/n-3 PUFA (i.e. changing a ratio from a high to a relatively lower one) on various diseases[18–24]. Recently, we have also developed another mouse model (FAT-2) to evaluate the effects of increased tissue levels of n-6 PUFA and a high tissue ratio of n-6/n-3 PUFA on the development of chronic diseases[11]. FAT-2 mice were engineered to expresses the FAT-2 gene from C. elegans and are capable of converting monounsaturated fatty acids (MUFA) into n-6 PUFA[11]. Therefore, FAT-2 mice have increased tissue levels of n-6 PUFA and a high tissue ratio of n-6/n-3 PUFA[11]. With the availability of FAT-1 and FAT-2 mice and crossbreeding, we were able to create a compound transgenic mouse model, namely the FAT-1+2 mouse, carrying both FAT-1 and FAT-2 genes that can endogenously synthesize both n-6 and n-3 PUFA[11]. Therefore, FAT-1+2 mice exhibit high tissue levels of both omega-6 and omega-3 fatty acids, with a balanced ratio of close to 1:1. Consequently, we have four genotypes of mice for use: wild type (incapable of producing essential fatty acids), FAT-1 (producing n-3 fatty acids), FAT-2 (producing only n-6 fatty acids), and FAT-1+2 (producing both n-6 and n-3 fatty acids)[11]. These mice exhibit four distinct PUFA phenotypes varying in the quantity of PUFA and n-6/n-3 ratio, even though they are all fed an identical diet with no need for dietary supplementation with corresponding PUFA[11]. Thus, use of these transgenic mice allows us to evaluate the authentic effects of different quantities and ratios of PUFA without the confounding factors of diet[6,11].

In the present study, we used these unique transgenic mouse models in combination with multi-omics technologies to determine the impact of varying amounts of omega-6 and omega-3 PUFA and their ratio on metabolic conditions and chronic disease development. We found that mice with varying n-6/n-3 PUFA ratios resulted in distinct gut microbiota, fecal and serum metabolites, and susceptibilities to cancer and certain metabolic disorders. FAT-2 transgenic mice with elevated n-6 PUFA levels and the highest n-6/n-3 PUFA ratios showed the most unfavorable metabolic conditions and the highest rate of liver cancer. These adverse health outcomes were largely prevented in FAT-1 and FAT-1+2 mice, which can convert n-6 PUFA to n-3 PUFA and have a balanced n-6/n-3 PUFA ratio. Our multi-omics study of host–microbiome interactions therefore uncovers differential health impacts of n-6 PUFA and n-3 PUFA and suggests an important role for a low tissue n-6/n-3 PUFA ratio in reducing the risk of chronic diseases.

## Results

**n-6/n-3 ratio influences the development of chronic disease.** The PUFA phenotypes of the four groups of mice used in this study (wild type, FAT-2, FAT-1, and FAT-1+2 mice) have been reported previously[11]. The tissue n-6 and n-3 PUFA content in these mice was further validated and shown in Supplementary Fig. 1. Although the levels of total PUFA in the tissues are the same between wild type and FAT-1 and between FAT-2 and FAT-1+2, the n-6/n-3 PUFA ratio is higher in the wild type and FAT-2 groups compared to FAT-1 and FAT-1+2 groups, respectively (Supplementary Fig. 1d, e, l). Specifically, the FAT-2 mice had the highest tissue n-6/n-3 PUFA ratio (Supplementary Fig. 1d, p), followed by the wild type mice, with the FAT-1 and FAT-1+2 mice having the lowest and most balanced n-6/n-3 PUFA ratios (Supplementary Fig. 1d, p). The FAT-1 and FAT-1+2 groups had a similar n-6/n-3 PUFA ratio, while the FAT-1+2 group had a higher quantity of PUFA compared to the FAT-1 group (Supplementary Fig. 1d, e, l).

We examined whether the four genotypes differed in the development of metabolic disorders, including metabolic endotoxemia, systemic inflammation, obesity, fatty liver, glucose intolerance, and cancer. The body weight of adult mice at the age of 8 months differed between the genotypes, with the highest body weights for the FAT-2 group and the lowest body weights for the FAT-1 and FAT-1+2 groups (Fig. 1a), while the food intake was not different between the groups (Supplementary Fig. 2a). Usually, the difference in fat mass observed during the young age would diminish when they become older. We thus followed up the phenotype of these mice for a prolonged Western diet exposure. Even after 16 months, the FAT-2 group maintained the highest body weight and the FAT-1+2 maintained the lowest body weight (Supplementary Fig. 2b). Increased body weight was correlated with increases in body fat mass, especially abdominal fat (Fig. 1b, c) in the 16-month-old mice. To assess the cause of this persisting body weight difference between groups, especially between FAT-2 and FAT-1+2, we investigated energy metabolism at the age of 16 months. As shown in Supplementary Fig. 2, FAT-2 mice spent less energy than FAT-1+2 and FAT-1 mice in the dark and light phases (Supplementary Fig. 2d, e), together with lower $CO_2$ production (Supplementary Fig. 2f, g) and $O_2$ consumption (Supplementary Fig. 2h, i). The respiratory exchange ratio (RER) was unaltered (Supplementary Fig. 2j, k). Moreover, it is notable that this difference in the energy expenditure was not due to a difference in physical activity (Supplementary Fig. 2l, m). Furthermore, morphological analyses of the adipose tissue showed that the size of adipocytes was much larger in white adipose tissue from the FAT-2 and wild type

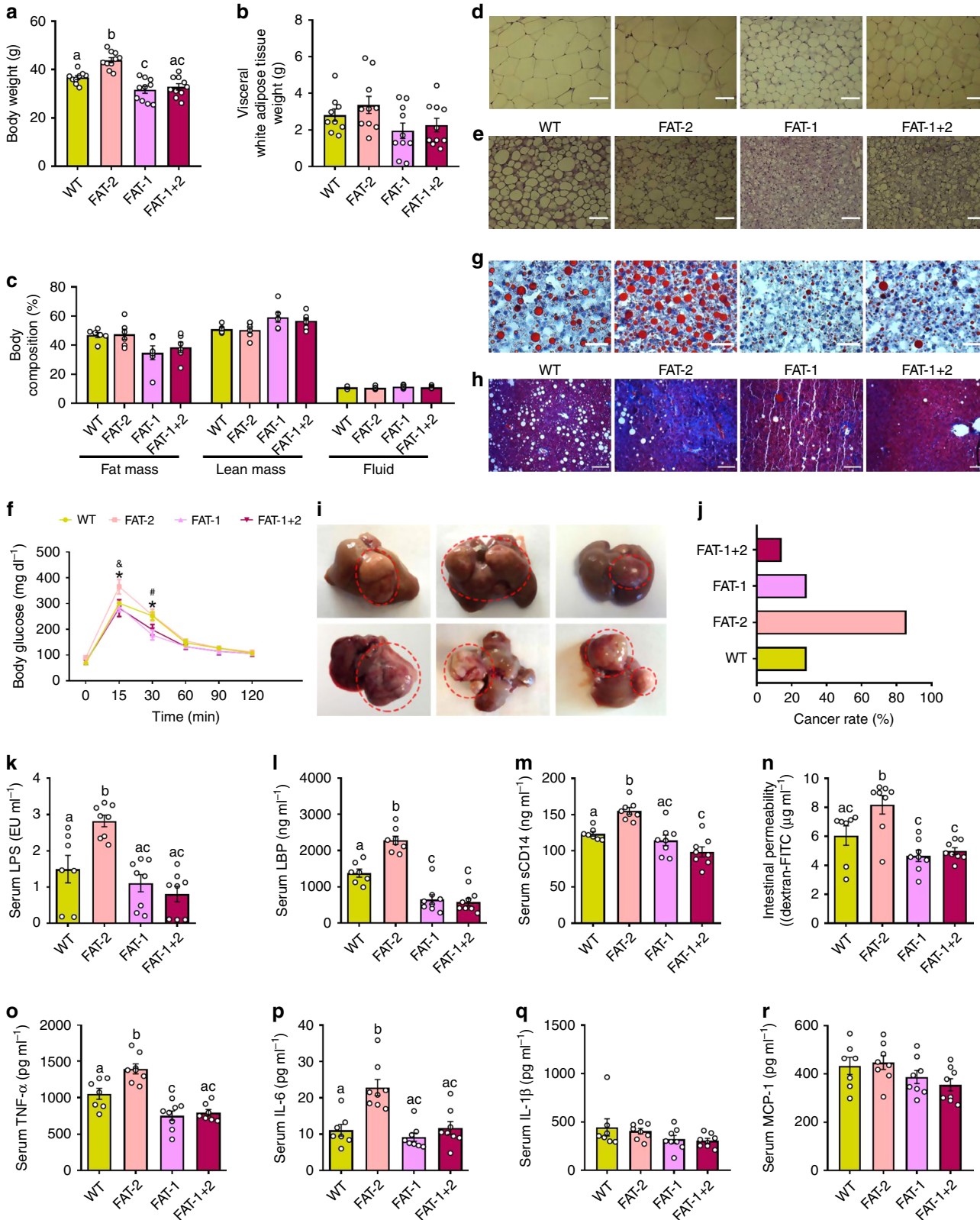

groups (Fig. 1d), and whitening of brown fat (an increase in large adipocytes) was observed in the brown adipose tissue of the FAT-2 and wild type mice, compared to the FAT-1 and FAT-1+2 groups (Fig. 1e). Likewise, glucose tolerance testing showed that FAT-1 and FAT-1+2 mice exhibited better glucose tolerance than the wild type and FAT-2 mice (Fig. 1f, Supplementary Fig. 3a). Oil red O staining revealed striking differences in lipid

accumulation among the four groups; specifically, the FAT-2 mice exhibited more lipid accumulation (5+ grade based on the number and size of stained fat droplets) and fatty liver development compared to their wild-type littermates (3+ grade), while lipid accumulation was markedly reduced in the FAT-1 and FAT-1+2 mice (1+ grade) (Fig. 1g). Furthermore, estimation of hepatic fibrosis showed a dramatically higher collagen deposition

**Fig. 1** Tissue omega-6/omega-3 PUFA ratio influences the development of metabolic disorders and cancer. Wild type (WT) and FAT-2, FAT-1, and FAT-1 +2 transgenic mice fed an identical Western diet for 16 months were subjected to several types of analyses at different time points. **a** Body weight at the age of 8 months. **b** Visceral white adipose tissue weight (g) after sacrificing the mice. **c** Nuclear magnetic resonance technique-based body composition analysis at the age of 16 months. Histopathological analysis (hematoxylin & eosin staining) of subcutaneous white adipose tissue (**d**) and inter-scapular brown adipose tissue (**e**) after sacrificing the mice. **f** Oral glucose tolerance test results obtained at the age of 8 months. **g** Fatty liver analysis performed on Oil Red O-stained liver specimens after sacrificing the mice. **h** Masson's trichrome staining (red, keratin and muscle fibers; blue, collagen; light red or pink, cytoplasm and dark brown to black, cell nuclei) performed on liver specimens to estimate the extent of fibrosis. **i** Anatomical shape and gross appearance of the livers with tumors from the FAT-2 mice. **j** Differences in the incidence rate of liver cancer between wild type, FAT-2, FAT-1, and FAT-1+2 mice. Markers of metabolic endotoxemia [lipopolysaccharides (LPS) (**k**), LPS-binding protein (LBP) (**l**), soluble CD14 (**m**) and intestinal permeability (serum levels of FITC-dextran macromolecules) (**n**)] and chronic low-grade inflammation [tumor necrosis factor-α (TNF-α) (**o**), interleukin-6 (IL-6) (**p**), IL-1β (**q**), and monocyte chemoattractant protein 1 (MCP-1) (**r**)] measured at the age of 12 months. For **c**, $n = 6$ per group. For **f**, **k–n** and **p–r**, wild type ($n = 7$), FAT-2, FAT-1, and FAT-1+2 ($n = 8$ per group). For **o**, wild type ($n = 7$), FAT-2 and FAT-1 ($n = 8$ per group), FAT-1+2 ($n = 7$ per group). For others, wild type ($n = 9$), FAT-2, FAT-1 and FAT-1+2 ($n = 10$ per group). Data shown as mean ± standard error of mean. Data with different superscript letters are significantly different ($P < 0.05$) according to ordinary one-way (**a–c**, **k–m**, **o–r**) or repeated measures two-way ANOVA (**f**) or Kruskal–Wallis test (**n**) followed by Tukey's or Dunn's multiple comparisons test. *FAT-2 vs. FAT-1 and FAT-2 vs. FAT-1+2; &wild type vs. FAT-2; #wild type vs. FAT-1. Scale bar for images **d** and **e**: 1000 μm; **g**: 2000μm; **h**: 3000 μm

(cirrhosis) in the FAT-2 liver specimens compared to the FAT-1 and FAT-1+2 groups (Fig. 1h). Strikingly, when the animals were sacrificed at the age of 16 months, we discovered that most of the FAT-2 mice (86%) had developed liver tumors (nodules and irregularities), while only very few mice from each of the other groups had liver tumors (Fig. 1i, j). Collectively, these findings indicate an increased risk for metabolic disorders and cancer in the groups with a high n-6/n-3 PUFA ratio, especially the FAT-2 mice that have the highest tissue level of n-6 PUFA, compared to the FAT-1 and FAT-1+2 groups with a balanced (lower) n-6/n-3 PUFA ratio.

Chronic low-grade inflammation is an underlying factor of many chronic diseases and closely associated with gut dysbiosis and metabolic endotoxemia[16]. We found that markers of metabolic endotoxemia (serum LPS, LBP, and sCD14, and intestinal permeability) (Fig. 1k–n) and chronic low-grade inflammation (TNF-a, IL-6, IL-1b, and MCP-1) (Fig. 1o–r) were increased in the FAT-2 mice compared to the wild-type mice, while these markers were reduced in the FAT-1 and FAT-1+2 mice. Furthermore, immunohistological staining showed that the expression of Toll-like receptor-4 (TLR4), a key regulator of the inflammatory pathway, was upregulated in the ileum, liver, and epididymal adipose tissue of the FAT-2 mice (Supplementary Fig. 3b–d), while expression of ZO-1, a tight junction protein, was downregulated in ileal tissue of the FAT-2 mice, compared to the other groups (Supplementary Fig. 3e), consistent with the increased metabolic endotoxemia and inflammation observed in the FAT-2 mice.

**n-6/n-3 PUFA ratio alters the composition of gut microbiota**. To examine the relationship between these divergent tissue fatty acid profiles and the microbial communities hosted by mice of differing genotypes, we performed high-throughput metagenomic sequencing and metabolomics of fecal samples. With principal coordinate analysis (Fig. 2a, Supplementary Fig. 4a) and hierarchical clustering (Supplementary Fig. 4b) methods, we discovered a distinct clustering of global microbiota composition among wild type, FAT-2, FAT-1, and FAT-1+2 genotypes, with the FAT-1 and FAT-1+2 genotypes forming a single cluster at one end, the wild-type cluster in the middle, and the FAT-2 cluster on the opposite end of the primary ordination axis (PERMANOVA results showed differences between groups except between FAT-1 and FAT-1+2 groups). The distribution of operational taxonomic units (OTUs) of seven bacterial groups with the largest magnitudes also showed distinct separation between wild type, FAT-2, and FAT-1 plus FAT-1+2 groups (Fig. 2b). Notably, OTUs belonging to the Enterobacteriacea and

Verromicrobiaceae families were most abundant in FAT-2 mice, followed by wild-type mice, while OTUs from the Bifidobacteriacea, Desulfovibrionaceae, and Bacteroidaceae families were enriched in FAT-1 and FAT-1+2 mice. Furthermore, hierarchical clustering based on the relative abundance of representative OTUs ($P < 0.05$) separated the mice as well as the bacterial groups into two primary clusters (Fig. 2c), with wild type and FAT-2 samples forming two distinct clusters in one clade and FAT-1 and FAT-1+2 samples combined in another clade. Biomarker analysis showed that Proteobacteria were increased in FAT-2 mice compared to the wild-type group, Firmicutes and Bacteroidetes were more abundant in FAT-1 mice, and Deltaproteobacteria and Actinobacteria were enriched in FAT-1+2 mice (Fig. 2d, Supplementary Fig. 4c). Furthermore, comparisons at the phylum level showed that FAT-2 mice had the highest abundance of Proteobacteria and the lowest abundance of Bacteroidetes, compared with the other genotypes (Fig. 2e). Both FAT-2 and wild type had lower abundances of Actinobacteria than FAT-1 and FAT-1+2 (Fig. 2e). Notably, we found that there was a striking difference in the Enterobacteriacea and Bifidobacteriacea between wild type and FAT-2 mice versus FAT-1 and FAT-1+2 mice (Fig. 2f, g). Relative quantification of bacterial groups by quantitative PCR (qPCR) was in accordance with these findings (Supplementary Fig. 4d, e). There were no differences between groups in the α-diversity measures (Supplementary Fig. 4f–j). Together, these results indicate that the n-6/n-3 PUFA ratio can affect gut microbiota composition. In particular, FAT-2 mice with an increased n-6/n-3 ratio exhibit greater abundance of Enterobacteriacea and depleted Bifidobacteriacea; in contrast, FAT-1 and FAT-1+2 mice with a balanced (lower) n-6/n-3 ratio exhibit enriched Bifidobacteriacea and reduced Enterobacteriacea. These findings support the notion that n-6 and n-3 PUFA have opposing effects on the gut microbiota.

Furthermore, similar to the patterns shown above in gut microbial profiles, the abundance of genes involved in functional KEGG (Kyoto Encyclopedia of Genes and Genomes) pathways were distinctly clustered by genotype (PERMANOVA results showed differences between groups except between FAT-1 and FAT-1+2 groups) (Fig. 2h). Next, from the list of differentially expressed KEGG pathways (false discovery rate (FDR)-corrected $P < 0.05$) represented by the inferred genomic content, we selected those associated with metabolic syndrome (MS), inflammation, bacterial translocation, non-alcoholic fatty liver disease (NAFLD), and cancer (Supplementary Table 1) and then analyzed the abundance of these pathways among the different mouse genotypes. We found that these KEGG pathways were upregulated in the FAT-2 mice compared to wild type mice, and

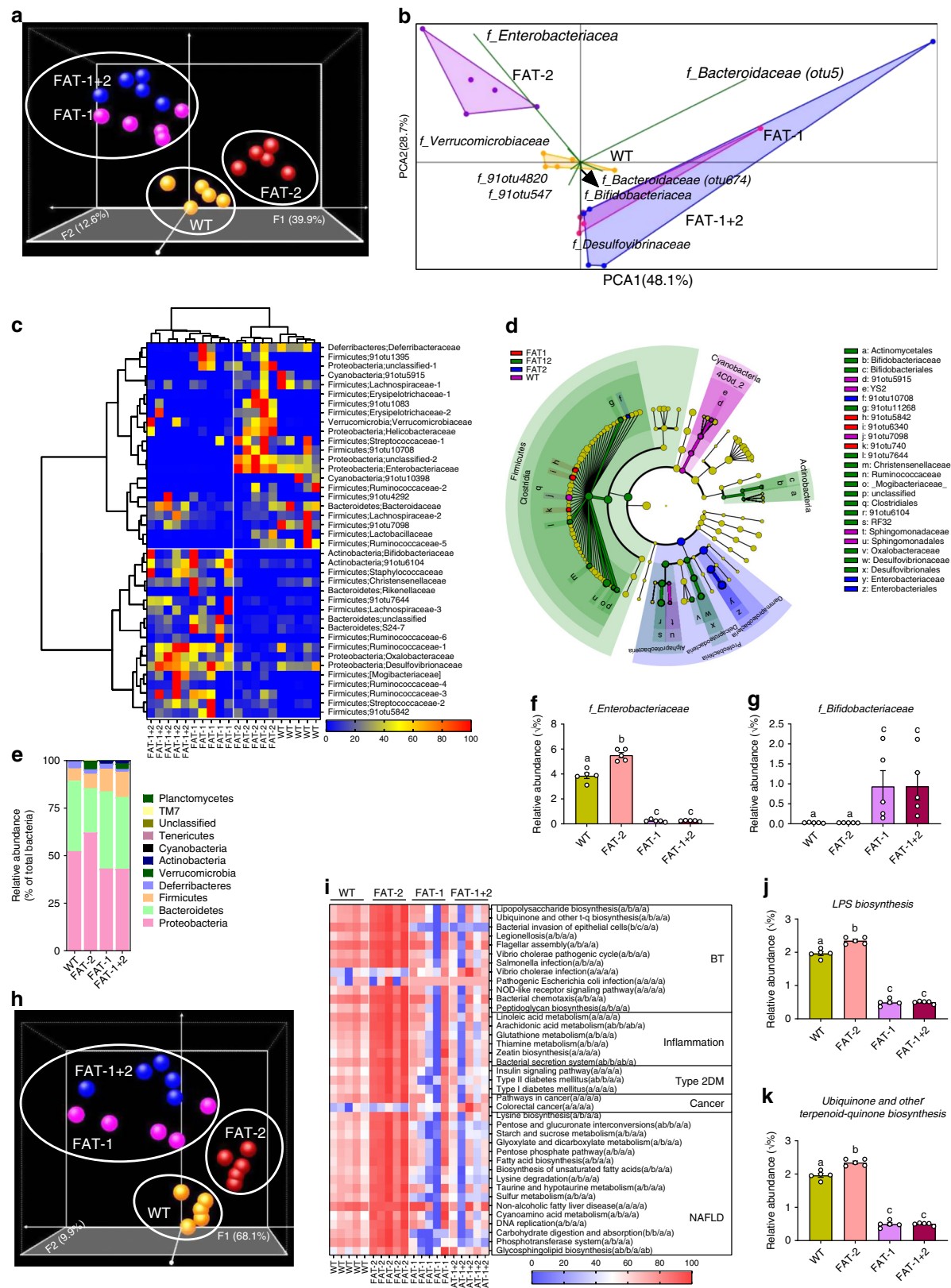

the level of these pathways was markedly reduced in the FAT-1/ FAT-1+2 groups (Fig. 2i–k). These findings further support that the changes in gut microbiota due to the n-6/n-3 ratio alteration may be associated with disease development.

**n-6/n-3 ratio alters fecal and serum metabolite profiles.** Next, we examined the potential differences in fecal metabolite profiles between the four mouse genotypes using an untargeted metabolomics approach together with partial least squares-discriminant

**Fig. 2** Alterations in the tissue-generated omega-6/omega-3 PUFA ratio impact the composition of fecal microbiota. Fecal microbiome analyses (V4 16S rRNA sequencing and predicted microbial functions) were performed on male wild type (WT), FAT-2, FAT-1, and FAT-1+2 transgenic mice ($n = 5$ per group) fed an identical Western diet for 12 months. **a** β-Diversity analysis performed on whole microbiota relative abundance using principal coordinate analysis (PCOA) with the Bray–Curtis dissimilarity index (BCD) followed by permutational multivariate analysis of variance (PERMANOVA) significance test. **b** Similarity percentage analysis with BCD was used to identify the specific genera with the greatest contribution to the differences observed between the groups, followed by principal component analysis (PCA) (variance–covariance type) showing the top eight operational taxonomic unit (OTU) scores included as vectors. The magnitude and direction correspond to the weights. **c** Hierarchical clustering with a heat map shows the relative abundance of representative OTUs (those with greatest difference between the four genotypes) group means (normalized to %) from each family selected for false discovery rate (FDR) corrected $P < 0.05$, obtained with differential abundance analysis. The OTUs are shown as phylum and family. **d** Cladogram generated from linear discriminant analysis (LDA) effect size showing the relationship between taxons (the levels represent, from the inner to outer rings, phylum, class, order, family, and genus). **e** Analysis at the phylum level using relative abundance (%). **f, g** Relative abundance ($Y = sqrt(Y)$ transformed) of family_Enterobacteriacea (E) and family_Bifidobacteriacea (%). **h** Relative abundance of predicted microbial genes related to metabolism was identified using PICRUSt analysis followed by PCOA analysis with BCD and PERMANOVA significance test. **i** Heat map shows the relative abundance (%) of representative predicted microbial genes (those with greatest difference between four genotypes) group means from each family selected for FDR-corrected $P < 0.05$, obtained with differential abundance analysis. BT, bacterial translocation; DM, diabetes mellitus; NAFLD, non-alcoholic fatty liver disease. **j, k** Relative abundance ($Y = sqrt(Y)$ transformed) of predicted bacterial genes involved in LPS biosynthesis and LPS biosynthesis proteins. Data shown as mean ± standard error of mean. $n = 5$/group. Data with different superscript letters are significantly different ($P < 0.05$) according to Mann–Whitney test (**f, g, j, k**)

analysis (PLS-DA). The relative abundance of fecal metabolites was distinctly clustered by the four genotypes (Fig. 3a). Abundance analysis of fecal metabolites linked with specific metabolic and pathological pathways showed distinct patterns among the four genotypes, as shown by the heat map in Fig. 3b. In particular, a number of metabolite markers of gut dysbiosis, inflammation and chronic disease (Supplementary Table 2) were elevated in FAT-2 mice and depleted in FAT-1 and FAT-1+2 mice (Fig. 3c–n). For example, levels of 1-methylnicotinamide (a marker of gut dysbiosis), cysteine and histidine (markers of increased intestinal permeability), lactate and spermidine (markers of intestinal inflammation), choline and trimethylamine N-oxide (TMAO) (markers of atherosclerosis), and gamma-glutymyl peptides (markers of liver injury) were relatively higher and indolepropionate (IPA) (which can reduce intestinal permeability) was lower in FAT-2 mice compared to the other genotypes (Fig. 3c–n). These results indicate that increased tissue n-6 PUFA content and n-6/n-3 PUFA ratio, as seen in the FAT-2 mice, could produce a fecal metabolite profile that promotes disease development. In addition, a correlation analysis found that the FAT-2 samples were clearly separated from the other three groups and clustered with increased Proteobacteria as well as microbial functional markers and fecal metabolite markers of disease pathways (Supplementary Fig. 4k). Conversely, the FAT-1 and FAT-1+2 samples were clustered on the opposite end of the axis with beneficial microbial families, including Actinobacteria (Supplementary Fig. 4k). Together, these findings indicate that the n-6/n-3 ratio is a key determinant of the gut microbiome and its functional pathways, as well as the fecal metabolite profile.

To examine whether changes in the n-6/n-3 ratio also affect the serum metabolite profile, we performed an untargeted metabolomics analysis on serum samples from the four genotypes of mice. PLS-DA showed that the four groups were separated according to both genotype and abundance of serum metabolites (Fig. 3o). Consistent with our findings on fecal metabolite profiles, serum metabolites known to be markers of inflammation, liver disease, metabolic syndrome, atherosclerosis, and liver cancer (Supplementary Table 3) were enriched in the FAT-2 mice compared to the wild-type group, and were markedly reduced in FAT-1 and FAT-1+2 mice (Fig. 3p, q). For example, serum levels of free arachidonic acid (marker of inflammation), ursodeoxycholate (UDCA) (marker of liver cancer), 12-hydroxyeicosatetraenoicacid (12-HETE) (marker of non-alcoholic steatohepatitis), orotidine (marker of liver injury), acyl carnitines (marker of insulin resistance), bilirubin (marker of type

2 diabetes), TMAO (marker of atherosclerosis), corticosterone (marker of obesity), and some other metabolites involved in aberrant amino acid biosynthesis, cell turnover regulation, reactive oxygen species neutralization, and eicosanoid pathways[25] were elevated in FAT-2 mice compared to the other genotypes (Fig. 3p, q, and Supplementary Fig 4l). These results further indicate that the increased tissue n-6 content and n-6/n-3 ratio can generate a serum metabolite profile that promotes the development of metabolic disorders.

**n-6/n-3 ratio influences microbe–metabolite interactions**. A functional relationship between the microbiome and metabolome is suggested by the similarity between OTU types and metabo-types[26]. The three data sets (microbiome, fecal, and serum metabolome) showed a high degree of concordance that was statistically significant in Monte Carlo simulations with a P value of 0.02. Superimposed microbiome and metabolomics data were separated not only by n-6/n-3 PUFA status but also by OTU type and metabotype (Fig. 4a). PLS-DA integration of three data sets (Fig. 4b) discriminated OTUs and metabolites that were correlated with each other and with the different n-6/n-3 PUFA status. Next, an inter-omic network (Fig. 4c, d) constructed using microbes and metabolites showed 8334 statistically significant correlations (Spearman's non-parametric rank correlation coefficient; $P < 0.05$) between two microbes, two metabolites, or a microbe and metabolite. The top three OTUs (Enterobacteriacea, Desulfovibrionaceae, and Bifidobacterium) and fecal (pyruvate, n-3 EPA, and n-6 DPA) and serum (n-3 EPA, n-6 DPA, and n-6 adrenate) metabolites were found to have a high inter-omic centrality as shown in the first two largest modules of the network (Fig. 4e, f). Betweenness centrality, which signifies the "bottleneck nodes (Supplementary Fig. 5a) that are crucial to the communication within the network[27], further highlights that this network strategy yields relevant key microbes and metabolites altered by tissue n-6/n-3 PUFA ratio.

**n-6/n-3 PUFA ratio influences host–microbiome interactions**. RV coefficient (0.73; $P = 0.001$) showed an overall measure of association between the tissue n-6/n-3 ratio-induced metabolic changes, microbes, and metabolites. Network-based analytical approaches extricate complex host–microbe interactions[26,28]. Correlation analysis resulted in a correlation network (Fig. 5a) consists of 1089 edges, 237 nodes, and 4 largest modules (Fig. 5b–e) (biologically important elementary units[28]). It is

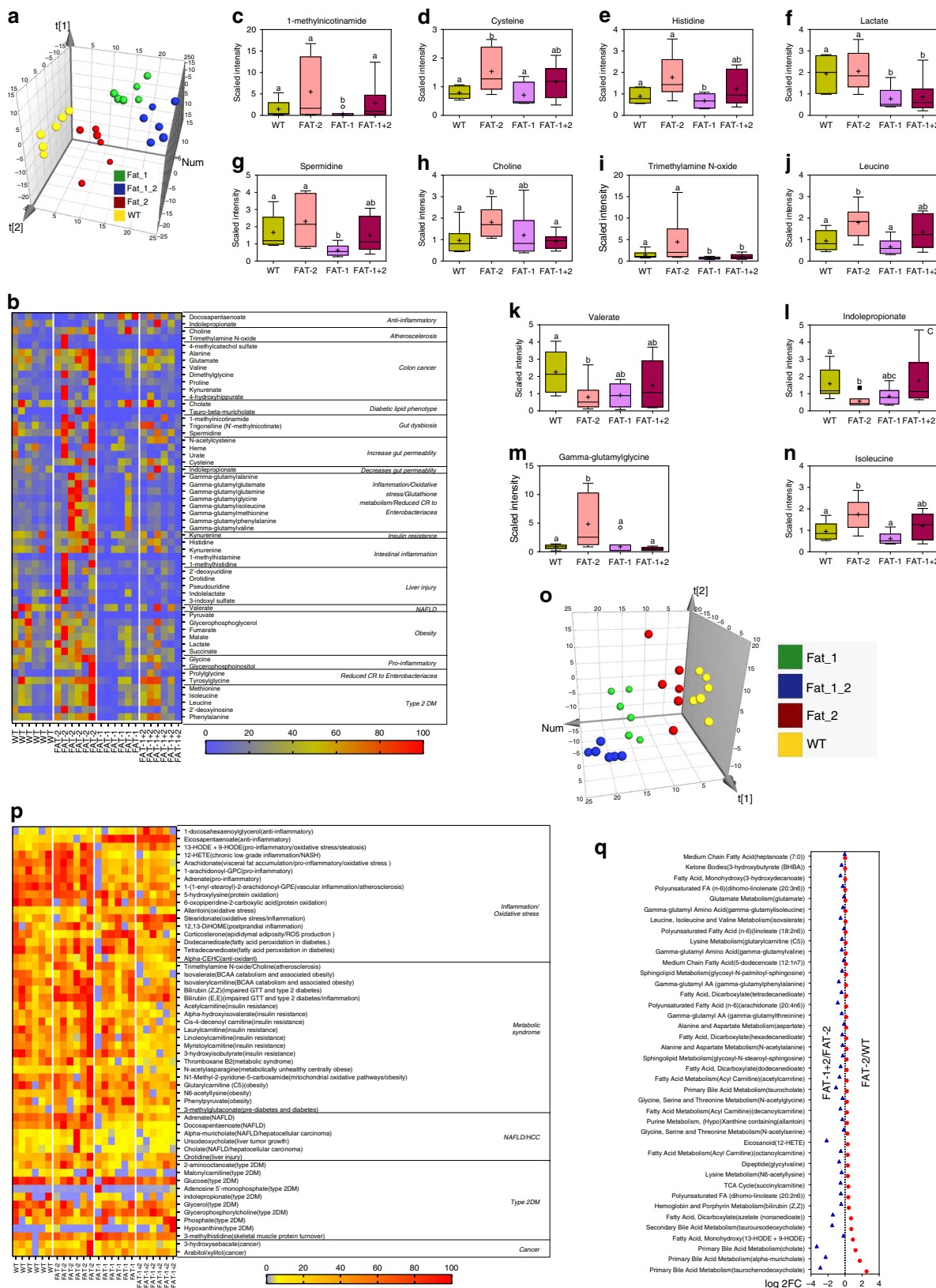

important to note that, according to the degree values, total n-6 PUFA (73), n-6/n-3 ratio (50), and intestinal permeability (IP) (52) in the first largest module, serum LPS (45) and LBP (43) and HCC (40) in the second module, serum IL-6 (37) and TNF-α (36) and body weight (34) in the third module, and total PUFA (14) and n-3 PUFA (27) in the fourth module were linked with

microbes and metabolites identified as biologically important with microbe–metabolite inter-omic analysis (Supplementary Fig. 5a). Next, parameters contributing to the multivariate PLS models were compared with the corresponding identified modules (Fig. 5b–e) in the correlation networks. A variability of 97%, 92%, 93%, 79%, 86%, 89%, and 82% for IP, LPS, LBP, HCC, IL-6,

**Fig. 3** Alterations in the tissue-generated omega-6/omega-3 PUFA ratio impact fecal and serum metabolite profiles. Global metabolic profiling on fecal (**a–n**) and serum (**o–q**) samples ($n = 6$/group) was performed on male wild type, FAT-2, FAT-1, and FAT-1+2 transgenic mice fed an identical Western diet for 14 months. **a** Three-dimensional (3D) view of score plots showing the results of supervised partial least squares-discriminant analysis with two-component model fitness parameters. To confirm the validation of the model, cross-validation analysis of variance (CV-ANOVA) with permutation tests ($n = 1000$) were performed.); $P < 0.05$ (CV-ANOVA); $R^2X$ (cum) $= 0.715$; $R^2Y$ (cum) $= 0.848$; $Q^2$(cum) $= 0.794$. **b** Heat map showing the abundance (normalized to percentage) of fecal metabolites with false discovery rate corrected $P < 0.05$. **c–n** Box-plots (box showing the mean, median, and the 25th and 75th percentiles, and the whiskers of the graph show the largest and smallest values) showing the abundance of selected fecal metabolites with FDR-corrected $P < 0.05$. **o** Partial least squares-discriminant analysis 3D plot with two-component model fitness parameters performed on global serum metabolite profile. To confirm the validation of the model, cross-validation analysis of variance with permutation tests ($n = 1000$) were performed; $P < 0.05$ (CV-ANOVA); $R^2X$ (cum) $= 0.706$; $R^2Y$ (cum) $= 0.716$; $Q^2$ (cum) $= 0.695$. **p** Heat map showing the abundance (normalized to percentage) of serum metabolites with false discovery rate corrected $P < 0.05$. **q** Log 2-fold change values of serum metabolites (involved in aberrant amino acid biosynthesis, cell turnover regulation, reactive oxygen species neutralization, and eicosanoid pathways) obtained with wild type vs. FAT-2 and FAT-2 vs. FAT-1+2 pairwise comparisons using Welch's two-sample t-test ($P < 0.05$). Data with different superscript letters are significantly different according to Welch's two-sample t-test ($P < 0.05$)

TNF-α, and body weight, respectively, were explained by the combination of key microbes and metabolites (Fig. 5f, Supplementary Data 1) altered by the tissue n-6/n-3 PUFA ratio.

Correlation network and principal component analysis (PCA) analyses on selected key parameters showed that the FAT-2 group was found to be closely associated with factors related to increased levels of n-6 PUFA metabolites, LPS production, gut permeability, inflammation, obesity, diabetes, fatty liver, and cancer, whereas FAT-1 and FAT-1+2 groups were associated with anti-inflammatory factors (Fig. 6a, b). Among the many correlations we found, the most important correlations are shown in Supplementary Fig. 5b–m. Notably, among the fecal PUFA tested, only n-6 PUFA were positively correlated with visceral adiposity (Supplementary Fig. 5lm). In summary, elevated tissue omega-6 PUFA status with an increased tissue n-6/n-3 PUFA ratio alters gut microbiota and gut microbial functional pathways, fecal metabolites production, and eventually microbiota-derived serum metabolites levels. These alterations plus adverse changes in the levels of serum non-microbiota-derived metabolites impacted directly by the elevated tissue n-6/n-3 ratio lead to increased intestinal permeability, metabolic endotoxemia, and chronic low-grade inflammation, resulting in the occurrence of chronic disease and cancer (Fig. 6c).

## Discussion

By using unique transgenic animal models and multi-omics technologies, our study has uncovered a potential pathway for the development of modern chronic diseases and cancer stemming from the dietary imbalance between n-6 and n-3 PUFA. Our discovery not only provides new insights into the etiology of chronic disease epidemics but also highlights the importance of balancing n-6 and n-3 PUFA in the diet to achieve a healthy tissue n-6/n-3 PUFA ratio as a key strategy for the management of chronic diseases. Our results emphasize that n-6 PUFA and n-3 PUFA are not equal, but actually exert differential or opposite effects on certain chronic health problems, indicating a need for reducing n-6 PUFA intake and increasing n-3 PUFA intake for improving health. This challenges many current dietary guidelines issued by governmental or health organizations, including the United States Dietary Guidelines (2015–2020), which recommends increasing intake of PUFA in general (mainly n-6 PUFA)[29].

We present a unique model for nutritional intervention studies to reliably elucidate the relationship between n-6/n-3 ratio and chronic disease by integrated multi-omics measurements of a series of inter-related biomarkers. Biomarkers including lipidome (profiles of fatty acids and their metabolites), gut microbiota, and metabolic endotoxemia (elevated Enterobacteriacea to *Bifidobacterium* ratio, serum LPS/LBP/gut permeability markers), inflammatory markers (e.g. TNF-a, IL-6, CRP), microbiota-derived metabolites (e.g. TMAO), host metabolomics, and pathological parameters can be accurately used for this model. Our unique genetic approach using transgenic mice allowed us to eliminate the confounding factors of diet (e.g. different types of diet[30] and the choice of control diet[26,31]) affecting gut microbiota composition and metabolite production[26]. Dietary modification is conventionally used to investigate the effects of different fatty acid profiles on gut microbiota and chronic disease development. However, this method is problematic since the diets used between study groups may contain not only different fatty acids but also variations in impurities, flavor, calories, or other components, as confounding factors that complicate interpretation of results[6,8,9]. Our model demonstrates the authentic effects of absolute amounts of n-6 and n-3 PUFA, total PUFA, and n-6/n-3 PUFA ratios in a single study. Given the fact that the Western diet consumption is associated with the imbalanced tissue n-6/n-3 ratio[32,33], our preclinical results provide evidence that balancing the n-6/n-3 ratio would be more important for better health outcomes. Whereas increasing the total PUFA consumption by only increasing the intake of n-6 PUFA resulted in adverse health outcomes with FAT-2 mice (Fig. 6c), this was not the case for FAT-1+2 mice because of their lower n-6/n-3 ratio, although both groups showed similar total PUFA. Likewise, increasing the total PUFA intake might not provide additional benefits when a balanced n-6/n-3 ratio is achieved because FAT-1 and FAT-1+2 mice showed similar healthy outcomes even though the FAT-1 mice have lower total PUFA than FAT-1+2. Similarly, although the total PUFA was the same between wild type and FAT-1 mice, lowering the n-6/n-3 ratio by elevating the n-3 PUFA (FAT-1 mice) resulted in better health outcomes. Overall, our results provide evidence that a balance between n-6 and n-3 PUFA is critical for good health and suggest that tissue n-6/n-3 ratio may be an important health biomarker. In this context, excessive intake of n-6 PUFA might be associated with adverse health outcomes and a simultaneous increase in n-3 PUFA consumption to balance the n-6/n-3 ratio is needed for good health.

In addition to the uniqueness of the mouse model, the other strengths of our study are highlighted as follows. Firstly, integrated multi-omic analyses is a powerful tool because the true power of our study design comes from the ability to examine results across the different omics levels to provide an integrated systems picture[26,34,35]. Secondly, network-based analytical approaches have the potential to help disentangle complex higher-order microbe–microbe, microbe–metabolite and microbe–host interactions, thereby broadening the applicability of microbiome research to personalized medicine and public health[28]. Finally, understanding host–microbe interactions is critical during times of disease, and balanced host–microbe interactions are necessary for maintaining homeostasis.

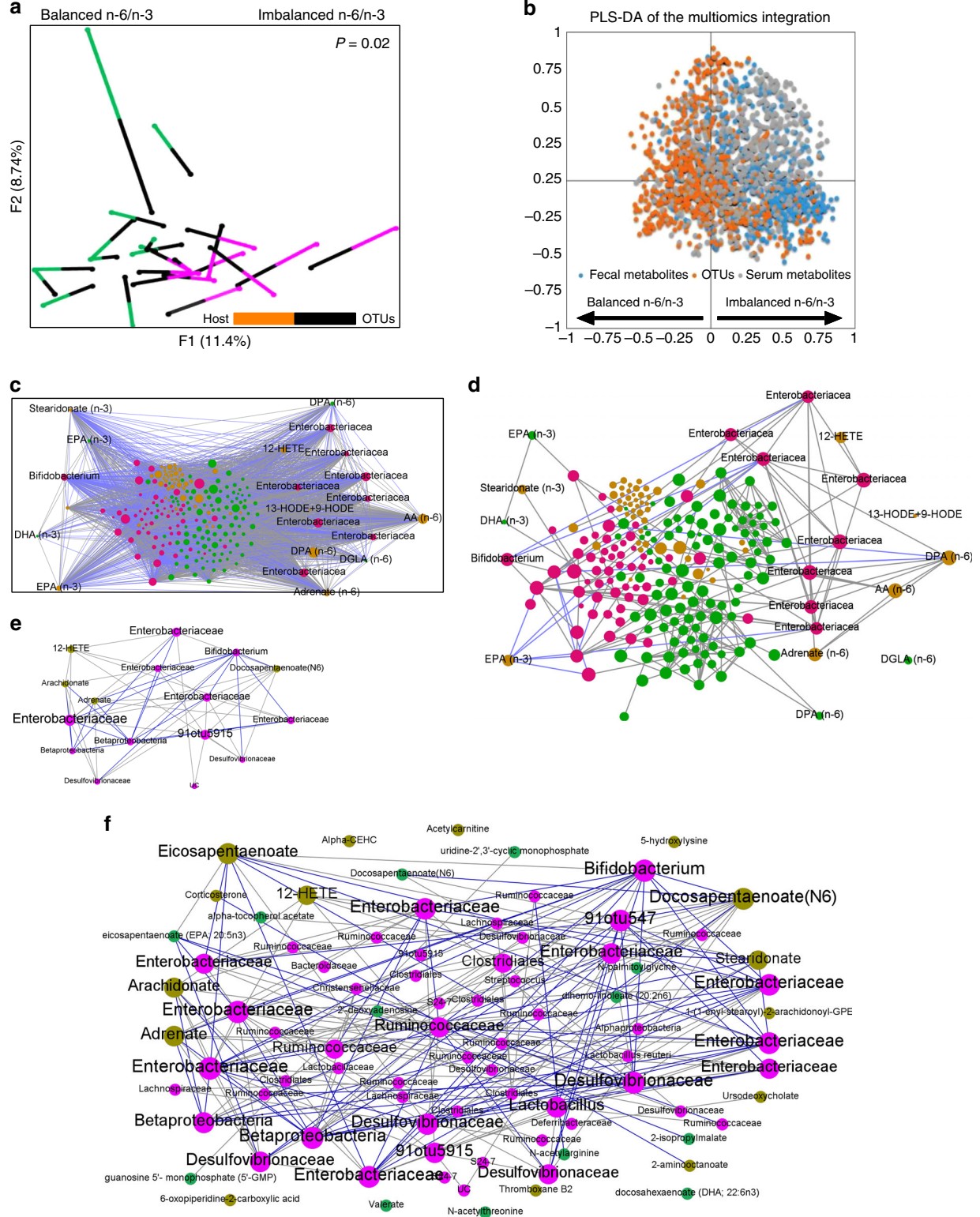

Our results showed a marked increase in LPS-producing and/or pro-inflammatory bacteria (e.g. Proteobacteria) and reductions in LPS-suppressing and/or anti-inflammatory bacteria (e.g. *Bifidobacterium*) observed in FAT-2 mice result in elevation of endotoxemia and inflammation. In support of this key observation, we found elevated microbial functions related to LPS biosynthesis and related proteins in the FAT-2 mice. Chronically altered changes in the gut–liver axis[36–38] and its role in the development of metabolic endotoxemia[39,40] and chronic low-grade inflammation[39,40] have been shown central to the development of chronic disease such as obesity, metabolic syndrome[41], and liver cancer[42]. Increased LPS production due to alterations in LPS-related gut bacteria and/or gut barrier function lead to the development of metabolic endotoxemia and associated chronic low-grade inflammation[16,37]. Recent studies have shown associations between markers of metabolic endotoxemia (LBP and sCD14) and systemic inflammation (TNF-α and IL-6) in subjects with obesity and metabolic syndrome[43,44]. Likewise, increasing

**Fig. 4** Inter-omic analysis reveals microbe–metabolite interactions between the tissue n-6/n-3 PUFA-associated microbial community type and metabotype. **a** Multiple factor analysis using Spearman type principal component analysis was performed to superimpose the microbiome ($n = 5$ per group) and fecal and serum metabolites ($n = 6$ per group) data (Monte Carlo simulations with a $P$ value equal to 0.02) associated with a balanced n-6/n-3 ratio (FAT-1/FAT-1+2 samples) and an imbalanced n-6/n-3 PUFA ratio (wild type/FAT-2 samples). Each line connects the microbial and metabolomics data from one sample. One end of each connecting line for an observation indicates the metabolites (differently colored to indicate the groups) and another end (black) indicates the microbiota [(operational taxonomic units (OTUs)]. **b** Multi-omics data integration showing partial leastsquares-discriminant analysis plot of all data (OTUs: 706; fecal metabolites: 554; serum metabolites: 554) for balanced n-6/n-3 (negative $x$ axis) versus imbalanced n-6/n-3 ratio (positive $x$ axis); $P = 0.003$ (CV-ANOVA); $R^2X = 0.792$; $R^2Y$ (cum) = 0.906; $Q^2$ (cum) = 0.837. **c**–**f** An inter-omic network was constructed with 225 nodes (filled circles) representing microbes (pink) and fecal (green) and serum (olive) metabolites with FDR-corrected $P$ values <0.05. Node size reflects inter-omic betweeness centrality — a measure of how many shortest paths within the entire network passes through the node in question (crucial to the communication within the network). Names of the selected microbes and metabolites with higher inter-omic degree centrality — the number of connections to nodes of the opposite data type (i.e., microbe–metabolite pairs) were shown. Edges represent statistically significant (spearman's non-parametric rank correlation coefficient) 5931 positive (gray) and 2403 negative (blue) correlations ($P < 0.05$) between microbe–microbe, metabolite–metabolite, or microbe–metabolite pairs. The entire network with 8334 edges showing all the correlations with $P < 0.05$ (**c**) and the entire network with edges showing only correlations having $R$ value > 0.8 for clarity of the inter-omic network (**d**). The first (**e**) and second (**f**) largest modules (biologically important elementary units of any biological network), which were separated from the full network according to the modularity scores

evidence suggests that the presence of endotoxemia is of substantial clinical relevance to patients with HCC[45]. In this context, decreasing the abundance of LPS-producing bacteria is a key mechanism for the reduction of metabolic endotoxemia, chronic low-grade inflammation, and the occurrence of chronic disease in the FAT-1 and FAT-1+2 mice with a balanced n-6/n-3 ratio. A recent study has shown that male FAT-1 mice fed high-fat diet for 6 weeks at young age showed a lean phenotype associated with higher energy expenditure than wild-type counterparts[46]. In the present study, the sustained increase in fat mass and eventually body weight observed in FAT-2 mice could be partially due to their lower energy expenditure, which may be another mechanism underlying the body weight gain in FAT-2 mice. The altered energy homeostasis (lower energy expenditure, $CO_2$ production and $O_2$ consumption) in the FAT-2 mice with elevated tissue n-6/n-3 ratio could be associated with impaired browning process[47] and altered intestinal endocannabinoid system[47]. However, this assumption warrants further investigation.

The altered gut microbiota-derived metabolites (GM-DMs) and their translocation to the liver and systemic circulation have been shown to have a role in metabolic syndrome[48] and hepatocellular carcinoma[25]. In our study, several fecal (e.g. indolepropionate) and serum (e.g. ursodeoxycholate) GM-DMs, which have been shown in the pathogenesis of metabolic syndrome and hepatocellular carcinoma (Supplementary Tables 2 and 3), were altered in the FAT-2 mice and these alterations were prevented in the FAT-1 and FAT-1+2 groups. Notably, increased abundance of TMAO-producing bacteria (e.g. Erysipelotrichaceae[49] and Enterobacteriacea[50]) was associated with elevated levels of fecal and serum TMAO (associated with insulin sensitivity[51], glucose metabolism[51], and atherosclerosis[52] and modified by fat inatke[51]) were observed in FAT-2 mice. In addition to LPS and GM-DMs, alterations in the non-microbiota host-derived metabolites may also play a role in the disease phenotypes of FAT-2 mice. A recent study identified 40 serum metabolites[25] that were elevated in hepatocellular carcinoma and cirrhosis subjects. Interestingly, the same 40 metabolites (e.g. UDCA[53] and 12-HETE[54]) were higher in the FAT-2 mice and lower in the FAT-1+2 mice. Taken together, unfavorable host–microbe interactions played a major role in the development of metabolic phenotypes and liver cancer due to imbalanced tissue n-6/n-3 ratios in the FAT-2 mice with the exact opposite situation occurring in the FAT-1 and FAT-1+2 mice. This conclusion is further supported by our high-throughput integrated multi-omic and host–microbiome interaction analyses. Modules derived from these analyses clearly showed a microbe–microbe, microbe–metabolite, and metabolite–metabolite interactions, and a strong association

between tissue n-6/n-3 ratio with LPS-related bacterial groups, several fecal and serum metabolites involved in chronic disease and markers of metabolic endotoxemia and chronic low-grade inflammation. This is important because a "module" in the network is an elementary unit of any biological network and biologically important when considered in isolation. Overall, our results indicate the necessity of having a balanced tissue n-6/n-3 ratio to create the balanced microbiome essential for the management of chronic disease.

The Western human diet has shifted dramatically in the last few decades from the diets that were consumed during most of human evolution. Key changes include increases in saturated fat, carbohydrates, and n-6 PUFA, and a decrease in n-3 PUFA[3,32]. As a result, many people today have an n-6 to n-3 PUFA ratio that favors n-6 PUFA by as much as 20:1. Historically, this ratio would have been closer to 1:1, and the discrepancy may contribute to modern health problems, including chronic diseases such as cancer. Our discovery that elevating tissue n-3 PUFA status and lowering the n-6/n-3 PUFA ratio can improve the gut microbiome profile, create a balanced host–microbiome interaction landscape, and suppress metabolic endotoxemia, and chronic low-grade inflammation provides two major implications for dealing with modern health problems. First, it supports the hypothesis that an excess of n-6 PUFA and deficiency of n-3 PUFA in the Western diet can contribute to modern chronic diseases (Fig. 6c). Second, it provides a new strategy for the prevention and treatment of chronic diseases by reducing the tissue n-6/n-3 ratio through n-3 PUFA supplementation and reducing n-6 PUFA intake. Further, the methodologies and results of our preclinical study emphasize the potential importance of reducing n-6 PUFA intake and/or increasing n-3 PUFA intake, rather than just increasing the total PUFA intake mainly through foods rich in n-6 PUFA.

In order to bridge the gap to clinical implementation, our results should be validated with future large-scale multi-omic analyses of the human gastrointestinal microbiome through nutritional intervention studies. Our work suggests that nutritional policies should be established that emphasize the differential effects of n-6 and n-3 PUFA rather than simply replacing saturated fatty acids with PUFA for the prevention of chronic disease. Finally, we acknowledge that a limitation of our study is the use of only male mice; future research in female mice is necessary because our findings obtained with male mice may not be translational to females.

In conclusion, integrated multi-omic analyses of our unique transgenic animal models uncover the tissue omega-6/omega-3 fatty acid imbalance as a critical risk factor for chronic disease.

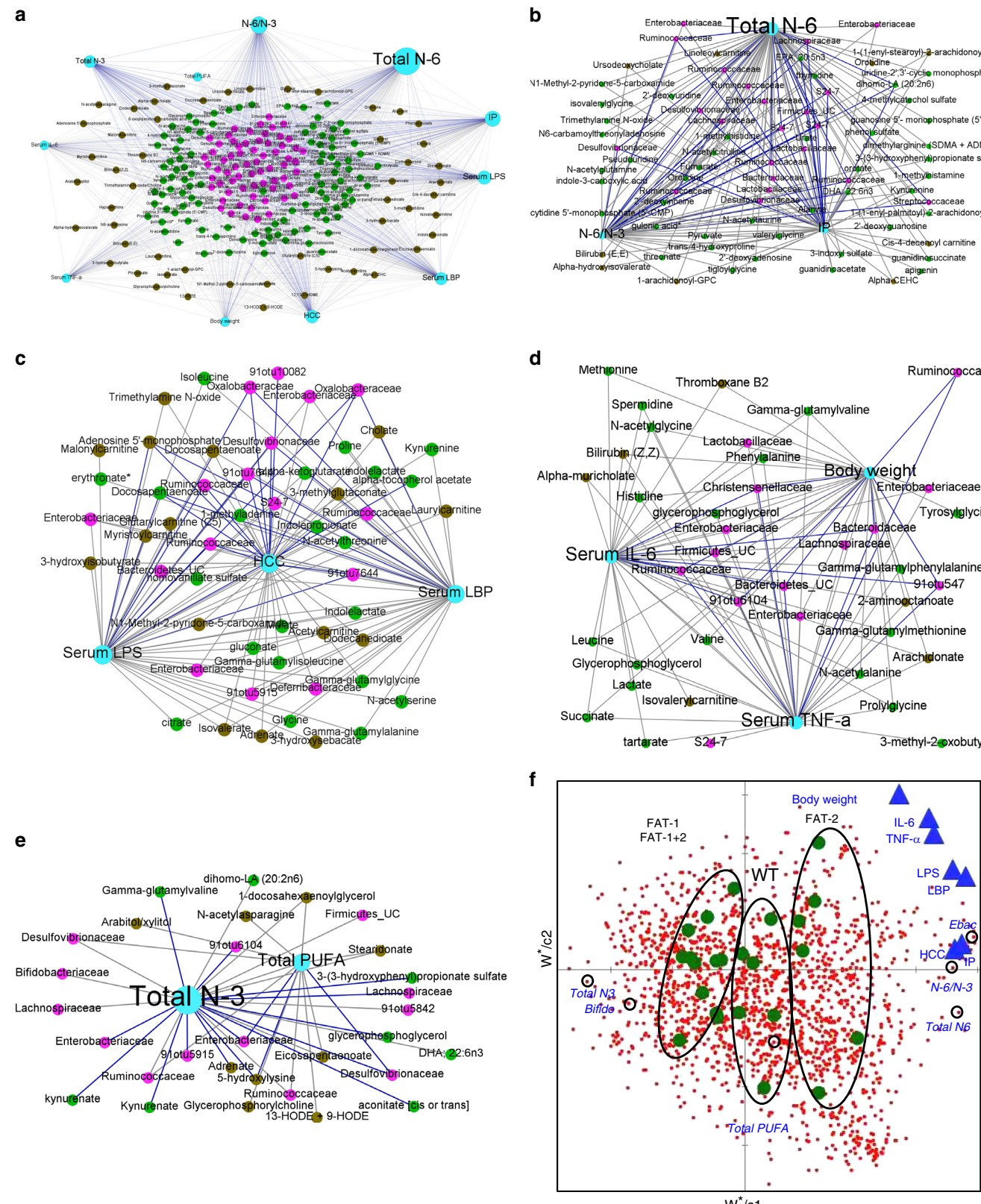

Decreased tissue n-6/n-3 ratio leads to beneficial changes in gut microbiota and gut microbial functional pathways. Subsequently, the positive alterations in fecal metabolite production and microbiota-derived and non-microbiota-derived serum metabolites suppress the development of metabolic endotoxemia and chronic low-grade inflammation and thereby reduce the risks for chronic diseases and cancer. Overall, this study demonstrates the importance of a low tissue omega-6/omega-3 PUFA ratio for maintaining good health and for the management of chronic diseases and cancer.

**Fig. 5** Network interactions uncover host–microbiome interactions driven by tissue n-6/n-3 PUFA status. **a–e** Host–microbiota interaction network built from Spearman's non-parametric rank correlation coefficient ($P < 0.05$) between 11 host parameters (serum total PUFA and n-6 and n-3 PUFA, n-6/n-3 ratio, body weight, HCC incidence, IP, serum LPS, LBP, TNF-α and IL-6) ($n = 6$ per group) and 66 microbial ($n = 5$ per group) and 159 metabolite parameters ($n = 6$ per group) with FDR-corrected $P$ values <0.05. Two hundred and thirty-seven nodes (filled circles) represents host parameters (cyan), microbes (pink), fecal (green), and serum (olive) metabolites. Node size reflects betweeness centrality — a measure of how many shortest paths within the entire network passes through the node in question (crucial to the communication within the network). In total, 1089 lines (edges) represent statistically significant correlations ($P < 0.05$) and are colored gray for 779 positive and blue for 310 negative correlations. The full network (**a**) with edges showing all the correlations and the four (**b–e**) largest modules (biologically important elementary units of any biological network), which were separated from the full network according to the modularity scores. **f** Partial least square (PLS)-regression loading score plot illustrating the association between host parameters (dependent variables — Y; blue triangles) and serum PUFA and microbial and metabolite parameters (explanatory variables — X; red dots). Explanatory variables of interest with variable importance in the projection (VIP) scores >1 were labeled with circles on the red dots. Samples from four different groups (wild type/FAT-2/FAT-1/FAT-1+2) were observed (green dots) and grouped using circles based on where they clustered on the plot. Leave-one-out cross-validation (LOO-CV) was applied. Ebac, Enterobacteriacea; Bifido, Bifidobacteriacea; HCC, hepatocellular carcinoma; IP, intestinal permeability; LPS, lipopolysaccharides; LBP, LPS-binding protein

## Methods

**Generation of transgenic mice.** As we described previousy[11], we generated a novel transgenic mouse model that can endogenously synthesize all essential fatty acids. Our strategy was to first create a FAT-2 transgenic mouse, possessing the *C. elegans FAT-2* gene encoding an enzyme that converts monounsaturated fatty acids (MUFA) into n-6 linoleic acid (LA), and then cross the FAT-2 transgenic mice with FAT-1 transgenic mice, which we generated previously to possess the *C. elegans FAT-1* gene, encoding an enzyme that converts n-6 to n-3 PUFA[10]. Through this procedure, we generated a compound FAT-1+2 transgenic mouse (Omega mouse) — that is capable of producing both n-6 and n-3 PUFA from a diet containing only saturated fat.

As described in our previous study[11], genotyping was carried out by removing the tip of the tail to acquire a DNA sample for reverse transcriptase (RT)-PCR, which was performed with the following primers: *FAT-2* forward, GCGGCCA GACCCAGACCATC; and *FAT-2* reverse, GGGCGAC GTGACCGTTGGTA. PCR products were run through gel electrophoresis on 2% agarose gel. Phenotyping by fatty acid composition analysis using gas chromatography (GC) was performed as previously described[55]. Tissue samples were ground to powder under liquid nitrogen and total lipids were extracted using chloroform/methanol (2:1, v/v). Fatty acids were then methylated by heating them at 100°C for 1 h under 14% boron trifluoride (BF3)-methanol reagent (Sigma-Aldrich, St. Louis, MO) and hexane (Sigma-Aldrich). Fatty acid methyl esters were analyzed by GC using a fully automated 6890N Network GC System (Agilent Technologies, Santa Clara, CA) equipped with a flame-ionization detector and an Omegawax 250 capillary column (30 m × 60.25 mm ID). Fatty acid standards (Nuchek Prep, Elysian, MN) were used to identify peaks of resolved fatty acids, and area percentages for all resolved peaks were analyzed using GC ChemStation Software (Agilent). The fatty acid C23:0 (20 mg per sample) was used as an internal standard to calculate the amount of each fatty acid measured. After identifying the genotype and phenotype, the FAT-2 mice were mated with wild-type C57BL6 mice to create the F1 generation. The F1 generation was then backcrossed with wild-type C57BL6 mice at least five times in order to verify that the gene is transmittable as well as to establish a pure background, so that FAT-2 lines could be maintained with a significant phenotype. Each generation was subjected to genotyping by RT-PCR and phenotyping by GC. The compound FAT-1+2 transgenic mice were created by crossbreeding heterozygous FAT-2 transgenic mice with heterozygous FAT-1 transgenic mice, which were previously generated by our group[10]. Genotyping by RT-PCR of the Omega mice was carried out with the following primers: *FAT-1* forward, TGTTCATGCCTTCT TCTTTTTCC; *FAT-1* reverse, GCGACCATACC TCAAACTTGGA; *FAT-2* forward, GCGGCCA GACCCAGACCATC; *FAT-2* reverse, GGGCGAC GTGACCGTTGGTA. Phenotyping by fatty acid composition analysis using GC was performed as previously described[55].

Animals in this study were maintained in accordance with the guidelines prepared by the institutional animal care and use committee (IACUC) at MGH based on the Care And Use of Laboratory Animals of the Institute of Laboratory Resources, National Research Council [Department of Health, Education and Human Services, Publication 85e23 (National Institutes of Health), revised 1985]. All animal protocols were reviewed and approved by the IACUC at MGH. Animals were sacrificed by the Animal Veterinary Medical Association (AVMA)-approved protocol of i.p. injection of pentobarbital (200 mg kg$^{-1}$).

**Animal diets.** After weaning and until the end this study, wild type ($n = 9$), FAT-2, FAT-1, and FAT-1+2 ($n = 10$ per group) mice were fed an identical basic diet, which contained the amount of n-6 and n-3 PUFA required for mice that do not carry the *FAT-1* and/or *FAT-2* transgenes. This is a modified Western diet (modification of TestDiet®AIN-76A Semi-Purified Diet 58B0 with beef tallow, coconut oil, glucose and fructose; 34.1% kcal from carbohydrate, 14.9% kcal from protein, 51% kcal from fat, totaling an energy content of 4.85 kcal g$^{-1}$) ordered from Test Diet (catalog # 1816187-200) (St. Louis, MO, USA). Please refer to Supplementary Table 4 for the complete fatty acid composition details of the diet.

**Animal experiments.** Mice were housed in a biosafety level 2 room in hard top cages with three or two mice per cage. Mice were maintained in a temperature-controlled room (22–24°C) with a strictly followed 12-h light/12-h dark diurnal cycle with food and water ad libitum. Four-week-old male wild type (wild type), FAT-2, FAT-1, and FAT-1+2 mice ($n = 9$–10 per group; 2 or 3 mice per cage), which were bred at the MGH animal facility, were fed an identical modified Western diet for 16 months, with no additional PUFA supplementation. All four groups of mice were subjected to several kinds of analysis at different time points. Animal body weight was measured at different time points and food intake was measured weekly. After 16 months of follow-up, mice were placed into individual cages in the Oxymax Comprehensive Laboratory Animal Monitoring System (CLAMS, Columbus Instruments) for an additional 24 h (at 23°C with a 12 h light/12 h dark cycle) for indirect calorimetry measurements for long-term phenotyping and assessment of energy expenditure, as previously described in Marvyn et al.[56]. Food and water were available ad libitum. The Oxymax system is an open-circuit indirect calorimeter for lab animal research allowing the measurement of oxygen consumption (VO$_2$), respiratory exchange ratio, and activity levels of mice. VO$_2$ is a measure of the volume of oxygen used to convert energy substrate in to ATP. The respiratory exchange ratio is the ratio of carbondioxide production (VCO$_2$) divided by VO$_2$, and can be used to estimate the fuel source for energy production based on the difference in the number of oxygen molecules required for the oxidation of glucose versus fatty acids[56]. A respiratory exchange ratio of 0.7 indicates that fatty acids are the primary substrate for oxidative metabolism, while a respiratory exchange ratio of 1.0 indicates that carbohydrate is the primary energy substrate[56]. Activity was calculated by summing the $X$ axis movement counts associated with horizontal movement. Body composition (fat and lean mass and fluid weight) of each mouse was determined by dual-energy X-ray absorptiometry (DEXA) according to the manufacturer's instructions (GE Lunar PIXImus 2). All values (g) were normalized using each mouse body weight and body composition was expressed as percentages. Mice were fasted for 6 h during the light phase period and blood was collected from the facial vein unless otherwise specified. Mouse feces and blood were collected at different time points as mentioned below and then sacrificed after 6 h fasting. Mouse organs (intestine, adipose tissue, liver, spleen, bone, brain, testis, tail, etc.) were flash frozen using liquid nitrogen and then stored at −80°C for further analysis.

**Extraction and purification of DNA from fecal samples.** At the age of 12 months, bacterial genomic DNA was extracted from fresh stool samples (~100–180 mg) from wild type, FAT-2, FAT-1, and FAT-1+2 mice using the QIAamp DNA Stool Mini Kit (Qiagen, Valencia, CA), following the manufacturer's instructions[16]. In order to increase its effectiveness, the lysis temperature was increased to 95°C. The eluted DNA was treated with RNase, concentration was determined by absorbance at 260 nm (A260), and purity was estimated by determining the $A_{260}/A_{280}$ ratio with a Nanodrop spectrophotometer (Biotek, Winooski, VT). DNA samples were diluted to 30 ng μl$^{-1}$, and this concentration was confirmed using both spectrophotometry ($A_{260}$) and fluorometry (DNAQF-1KT; Sigma, USA).

**Genomic DNA library preparation.** Genomic DNA samples ($n = 5$ per group) were sent to the microbiome analysis company (Second Genome, Inc., CA, USA) to perform V4 16S rRNA gene sequencing. As described previously[57], to enrich the sample for bacterial 16S V4 rDNA region, DNA was amplified using fusion primers designed against the surrounding conserved regions which are tailed with sequences to incorporate Illumina (San Diego, CA) adapters and indexing bar-codes. Each sample was PCR amplified with two differently barcoded V4 fusion primers. Samples that met the post-PCR quantification minimum were advanced for pooling and sequencing. For each sample, amplified products were concentrated using a solid-phase reversible immobilization method for the purification of PCR products and quantified by qPCR.

**Microbiome profiling.** A pool containing 16S V4-enriched, amplified, barcoded samples was loaded into a MiSeq® reagent cartridge, and then onto the instrument

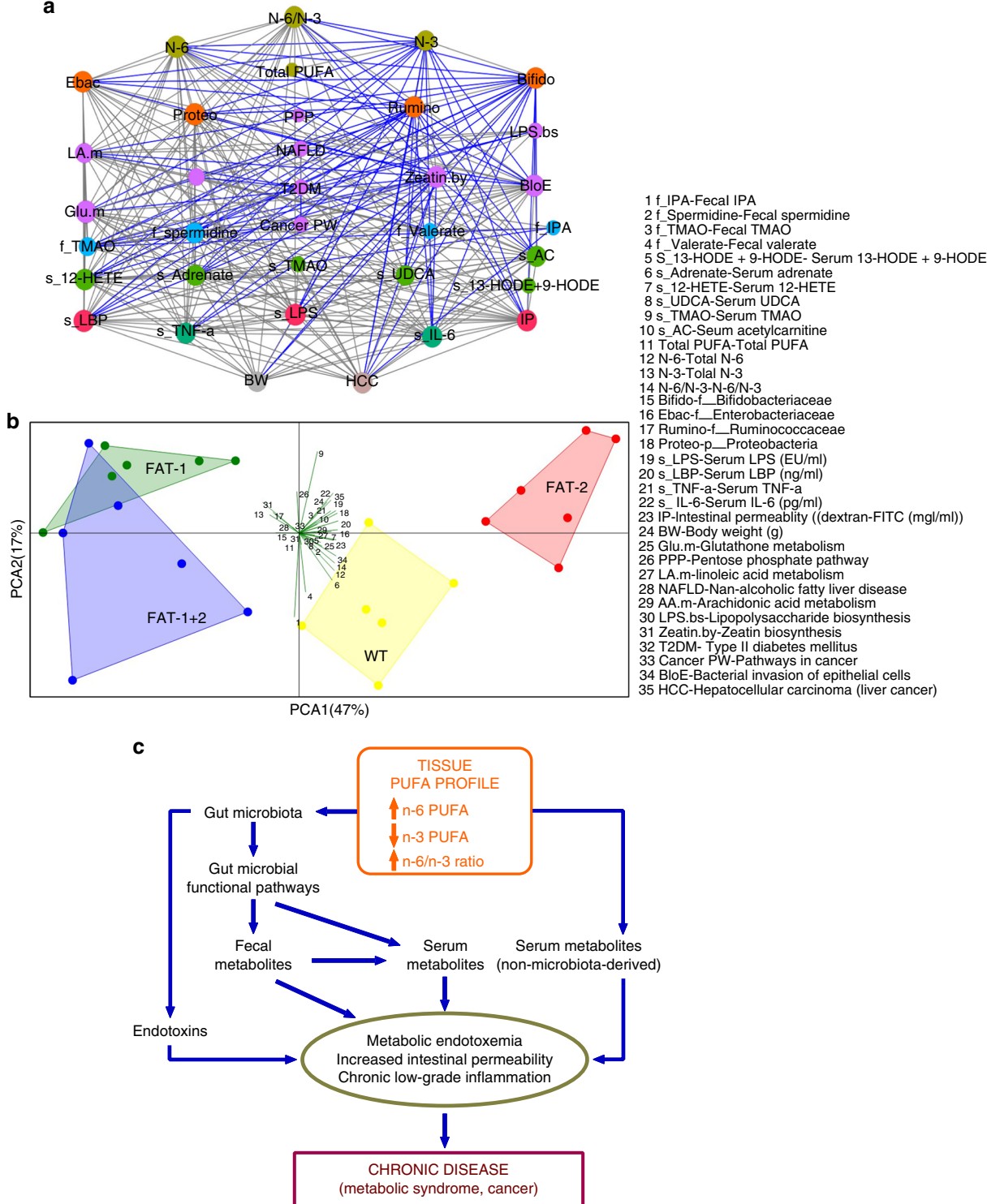

1 f_IPA-Fecal IPA
2 f_Spermidine-Fecal spermidine
3 f_TMAO-Fecal TMAO
4 f_Valerate-Fecal valerate
5 S_13-HODE + 9-HODE- Serum 13-HODE + 9-HODE
6 s_Adrenate-Serum adrenate
7 s_12-HETE-Serum 12-HETE
8 s_UDCA-Serum UDCA
9 s_TMAO-Serum TMAO
10 s_AC-Seum acetylcarnitine
11 Total PUFA-Total PUFA
12 N-6-Total N-6
13 N-3-Tolal N-3
14 N-6/N-3-N-6/N-3
15 Bifido-f__Bifidobacteriaceae
16 Ebac-f__Enterobacteriaceae
17 Rumino-f__Ruminococcaceae
18 Proteo-p__Proteobacteria
19 s_LPS-Serum LPS (EU/ml)
20 s_LBP-Serum LBP (ng/ml)
21 s_TNF-a-Serum TNF-a
22 s_ IL-6-Serum IL-6 (pg/ml)
23 IP-Intestinal permeablity ((dextran-FITC (mgl/ml))
24 BW-Body weight (g)
25 Glu.m-Glutathone metabolism
26 PPP-Pentose phosphate pathway
27 LA.m-linoleic acid metabolism
28 NAFLD-Nan-alcoholic fatty liver disease
29 AA.m-Arachidonic acid metabolism
30 LPS.bs-Lipopolysaccharide biosynthesis
31 Zeatin.by-Zeatin biosynthesis
32 T2DM- Type II diabetes mellitus
33 Cancer PW-Pathways in cancer
34 BloE-Bacterial invasion of epithelial cells
35 HCC-Hepatocellular carcinoma (liver cancer)

along with the flow cell. After cluster formation on the MiSeq instrument, the amplicons were sequenced for 250 cycles with custom primers designed for paired-end sequencing. Samples are processed in a Good Laboratory Practices (GLP) compliant service laboratory running Quality Management Systems for sample and data tracking. The laboratory implements detailed standard operating procedures (SOPs), equipment and process validation, training, audits, and document control measures. Quality control (QC) and assurance (QA) metrics were maintained for all samples.

**Microbiome data analysis.** The full data analysis pipeline for Second Genome's Microbial Profiling Service incorporates several separate stages: pre-processing, summarization, normalization, alpha-diversity metrics (within sample diversity),

beta-diversity metrics (sample-to-sample similarity), ordination/clustering, sample classification, and significance testing. The report was generated using second-genomeR package: 0.2.4.

For OTU selection, paired-end reads were merged, quality filtered, and de-replicated with USEARCH[58]. Resulting unique sequences were then clustered at 97% similarity by UPARSE (de novo OTU clustering) and a representative consensus sequence per de novo OTU was determined. The clustering algorithm also performs chimera filtering to discard likely chimeric OTUs. Sequences that passed quality filtering were then mapped to a set of representative consensus sequences to generate an OTU abundance table. Representative OTU sequences were assigned taxonomic classification via mothur's Bayesian classifier trained against the Greengenes reference database of 16S rRNA gene sequences clustered at 99%. After taxa were identified for inclusion in the analysis, the values used for

**Fig. 6** A proposed mechanism (developed with integrated multi-omic analysis of transgenic animal models) showing the relationship between tissue omega-6/omega-3 fatty acid imbalance and the development of chronic disease. **a** To assemble the overall correlations among our data (*n* = 6/group), we performed correlation network analysis (Spearman's non-parametric rank correlation coefficient). Each node was colored according to the data type and sized based on the betweenness centrality, which quantifies the influence of a node in connecting other nodes in a network. Edges (lines) represent statistically significant correlations, and are colored light black for positive and blue for negative correlations. Please refer panel **b** for abbreviations. **b** PCA (correlation type) analysis was performed on selected key parameters (*n* = 6/group), including the genotypes, gut microbiota, fecal and serum metabolites, and markers of metabolic disorders. **c** Diagram illustrating our proposed mechanism of tissue omega-6/omega-3 fatty acid imbalance leading to the development of chronic disease. Elevated tissue omega-6 PUFA status and an increased tissue n-6/n-3 fatty acid ratio alters gut microbiota and gut microbial functional pathways, which in turn adversely influence fecal metabolites production and eventually microbiota-derived serum metabolites levels. These alterations plus adverse changes in the levels of serum non-microbiota-derived metabolites directly influenced by the elevated tissue n-6/n-3 fatty acid ratio lead to increased intestinal permeability, development of metabolic endotoxemia and chronic low-grade inflammation, resulting in the occurrence of chronic disease (metabolic syndrome and cancer)

each taxon-sample intersection were populated with the abundance of reads assigned to each OTU in an "OTU table". A corresponding table of OTU Greengenes classification was generated as well.

*Alpha-diversity (within sample diversity) metric:* "Observed" diversity is the simply the sum of unique OTUs found in each sample, also known as sample richness. Chao1 calculates the estimated sample richness (number of OTUs) based on sequencing depth and taking into account rare taxa that may be present in a sample. Shannon diversity uses the richness of a sample along with the relative abundance of the present OTUs to calculate a diversity index.

*Beta-diversity (sample-to-sample dissimilarity) metrics:* All profiles were inter-compared in a pairwise fashion to determine a dissimilarity score and store it in a distance dissimilarity matrix. Distance functions produce low dissimilarity scores when comparing similar samples. Abundance-weighted sample pairwise differences were calculated using the Bray–Curtis dissimilarity — the ratio of the summed absolute differences in counts to the sum of abundances in the two samples. The binary dissimilarity values were calculated with the Jaccard index. This metric compares the number of mismatches (OTUs present in one but absent in the other) in two samples relative to the number of OTUs present in at least one of the samples.

*Ordination, clustering, and classification methods:* Two-dimensional ordinations and hierarchical clustering maps of the samples in the form of dendrograms were created to graphically summarize the inter-sample relationships. Principal Coordinate Analysis is a method of two-dimensional ordination plotting that is used to help visualize complex relationships between samples. Principal Coordinate Analysis uses the sample-to-sample dissimilarity values to position the points relative to each other by maximizing the linear correlation between the dissimilarity values and the plot distances. To create dendrograms, the samples from the distance matrix are clustered hierarchically using the ward method.

*Whole microbiome significance testing:* Permutational Analysis of Variance (PERMANOVA) was used for finding significant differences among discrete categorical or continuous variables. In this randomization/Monte Carlo permutation test, the samples are randomly reassigned to the various sample categories, and the between-category differences are compared to the true between-category differences. PERMANOVA utilizes the sample-to-sample distance matrix directly, not a derived ordination or clustering outcome.

*Taxon significance testing:* Univariate differential abundance of OTUs was tested using a negative binomial noise model for the overdispersion and Poisson process intrinsic to this data, as implemented in the DESeq2 package[59], and described for microbiome applications in ref. [60]. It takes into account both technical and biological variability between experimental conditions. DESeq was run under default settings and *q*-values were calculated with the Benjamini–Hochberg procedure to correct *P* values, controlling for false discovery rates.

**Identification of metagenomic biomarkers**. The SIMPER (Similarity Percentage analysis) method was applied to whole microbiome relative abundance data to identify the top 10 taxa by abundance. Their contribution to groups (between and within groups) was analyzed as previously described using the PCA variance–covariance type ordination (PAST version 3.11 software) method[37]. Differential abundance analysis (non-parametric ANOVA with Benjamini–Hochberg FDR-corrected *P* values<0.05) was performed on the relative abundance data at different levels of taxonomy to identify taxa with FDR-corrected *P* values <0.05 (XLSTAT software; Addinsoft, USA)[37]. Their relative abundance (normalized to percentage) was then shown by heat map with hierarchical clustering analysis[61] using GraphPad Prism version 8 (La Jolla, CA)[37]. Linear Discriminant Analysis (LDA) Effect Size (LEfSe) is a biomarker discovery and explanation tool for high-dimensional data. It couples statistical significance with biological consistency and effect size estimation[37,62]. Microbiome-based biomarker discovery was performed with LEfSe using the online galaxy server (https://huttenhower.sph.harvard.edu/galaxy/). LDA scores derived from the LEfSe analysis were used to show the relationship between taxa using a cladogram (circular hierarchical tree) of significantly increased or decreased bacterial taxa in

the gut microbiota between groups[37]. Levels of the cladogram (Fig. 2d) represent, from the inner to outer rings: phylum, class, order, family, and genus. Color codes indicate the groups, and letters indicate the taxa that contribute to the uniqueness of the corresponding groups at an LDA of >2.0. Three-dimensional (3D) views of principal coordinate analysis (PCoA) score plots were prepared using XLSTAT-3D Plot[37]. UniFrac is a distance metric used for comparing biological communities. It differs from dissimilarity measures such as Bray–Curtis dissimilarity in that it incorporates information on the relative relatedness of community members by incorporating phylogenetic distances between observed organisms in the computation. Weighted UniFrac PCoA analysis and LEfSe analysis to obtain LDA scores were performed on the whole microbiome relative abundance data using MicrobiomeAnalyst[63]. Generation of rare faction curves using whole microbiome OTU data and hierarchical clustering were performed using PAST version 3.11 (ref. [37]). The non-weighted group average (unweighted pair-group with arithmetic means, UPGMA) was used to perform hierarchical clustering analysis. The diagram based on the Bray–Curtis distance matrix was obtained using PAST version 3.11 (ref. [37]). Class tree was used to demonstrate similarity between samples, through the clustering tree branch length measure cluster effect. Fecal microbiome data analysis was also done using PAST version 3.11 software program[37] to make two-dimensional principal coordinate and principal component plots and alpha diversity indices and to perform correlation analysis. XLSTAT for Microsoft Excel (Addinsoft SARL, Paris, France)[37] was used to analyze differential expression and create a heat map (Fig. 2c) showing the relative abundance of representative OTUs selected for *P* < 0.05, obtained using a non-parametric differential expression test with the Benjamini–Hochberg procedure to correct *P* values between all four groups and then grouped into families. One representative OTU with the greatest difference between the group's means from each family was selected for inclusion in the heat map diagram[37]. OTUs are shown as**:** Phylum and Family. SIMCA 14.1 (Umetrics, Stockholm, Sweden)[64] software was used to make two-dimensional principal component plots and GraphPad Prism version 8 (GraphPad Software, La Jolla, CA) for other analyses. To determine the key genus profiles distinguishing genotypes, a Linear Discriminant Analysis Effect Size (LEfSe analysis)[37] was performed using the default parameters and the cladogram was generated accordingly.

**Microbiome functional analysis**. Phylogenetic Investigation of Communities by Reconstruction of Unobserved States (PICRUSt) is a well-documented tool designed to impute metagenomic information based on 16S input data[37]. PICRUSt analysis (*n* = 5 per group) was performed by Second Genome, Inc. The PICRUSt database was derived from 2590 genomes in v3.5 of IMG, which is 11% and 26% fewer than the genome numbers in recent Kyoto Encyclopedia of Genes and Genomes (KEGG 70.1) or BioCyc 18.0, respectively. Thus, an alternate approach was applied to leverage the most up-to-date genome database to infer metagenomes of 16S rRNA sequenced samples (Piphillin[65], Second Genome Inc.). Both the KEGG 70.1 and BioCyc 18.0 databases were used as reference genomes. A genome was inferred for each 16S rRNA OTU based on the sequence identity between the OTU's representative sequence and the nearest neighbor 16S rRNA sequence from the genome databases restricted to a minimum identity of 97%. OTU abundance was normalized by 16S rRNA copy numbers, and then multiplied by the gene contents of each inferred genome to predict each sample's metagenome. Abundance of microbial function related genes and KEGG pathways were identified and non-parametric analysis of differential expression (*P* < 0.05; Post hoc corrections: Benjamini–Hochberg) was done using XLSTAT software on all four groups of data[37]. Also, pairwise comparisons were also performed between groups using GraphPad Prism version 8 (multiple T-TESTS; one per row; *P* values computed without assuming consistent standard deviation (Welch's correction); false discovery rate (FDR) was set to 0.05 and the differences between groups were considered significant only FDR-corrected *P* < 0.05).

**Quantitative real-time PCR (qRT-PCR) assessment of fecal microbiota**. Targeted fecal microbial profiling was performed as previously described[37]. Briefly, qRT-PCR was performed with a PRISM 9000 Light Cycler (Applied Biosystems, USA) using the iTaq universal SYBR Green Supermix (Bio-Rad, USA) and group-

specific primers (Supplementary Table 5) for total bacteria, family Enterobacteriacea, genus *Escherichia* and genus *Bifidobacterium*. Samples ($n = 9$–10 per group) and controls were run in duplicate in total reaction volumes of 20 μl per well, containing 500 nM primers and 40 ng genomic DNA. Amplification and data acquisition was performed according to the protocol provided with SYBR Green (Bio-Rad, Hercules, CA). By subtracting the cyclic threshold (Ct) values of total bacteria from the Ct values of each bacterial group, we estimated and compared the relative quantification of a specific bacterial group.

**Global metabolic profiling by Metabolon Inc**. *Sample collection:* At 14 months of age, fecal materials were collected from wild type, FAT-2, FAT-1, and FAT-1+2 mice ($n = 6$ per group) using sterile 2 ml tubes, weighed, flash frozen in liquid nitrogen, and stored at −80°C until shipping. Whole blood was drawn from the facial vein in 6 h fasted mice, left at room temperature for 25–30 min, clear serum-separated by centrifugation (6000 RPM for 6 min), flash frozen in liquid nitrogen, and stored at −80°C until shipping. Consistency in sample handling was maintained by minimizing operational variation (collection technique, time of sampling, time to freezer, etc.). Both feces and serum samples were shipped to Metabolon Inc. (Durham, NC) where they were extracted and prepared for analysis using a previously described standard solvent extraction method[66].

*Sample accessioning*: Following receipt by Metabolon Inc., samples were inventoried and immediately stored at −80°C. Each sample received was accessioned into the Metabolon Laboratory Information Management System (LIMS system) and was assigned by the LIMS a unique identifier that was associated with the original source identifier only. This identifier was used to track all sample handling, tasks, results, etc. The samples (and all derived aliquots) were tracked by the LIMS system. All portions of any sample were automatically assigned their own unique identifiers by the LIMS when a new task was created; the relationship of these samples was also tracked. All samples were maintained at −80°C until processed.

*Sample preparation:* Samples were prepared using the automated MicroLab STAR® system from Hamilton Company (https://www.hamiltoncompany.com/). Several recovery standards were added prior to the first step in the extraction process for QC purposes. To remove protein, dissociate small molecules bound to protein or trapped in the precipitated protein matrix, and to recover chemically diverse metabolites, proteins were precipitated with methanol under vigorous shaking for 2 min (Glen Mills GenoGrinder 2000) followed by centrifugation. The resulting extract was divided into five fractions: two for analysis by two separate reverse phase (RP)/ultra-performance liquid chromatography–tandem mass spectrometry (UPLC-MS/MS) methods with positive ion mode electrospray ionization, one for analysis by RP/UPLC-MS/MS with negative ion mode electrospray ionization, one for analysis by HILIC/UPLC-MS/MS with negative ion mode electrospray ionization, and one sample was reserved for backup. Samples were placed briefly on a TurboVap® (Zymark) to remove the organic solvent. The sample extracts were stored overnight under nitrogen before preparation for analysis.

*QA/QC:* Several types of controls were analyzed in concert with the experimental samples: a pooled matrix sample generated by taking a small volume of each experimental sample (or alternatively, use of a pool of well-characterized human plasma) served as a technical replicate throughout the data set; extracted water samples served as process blanks; and a cocktail of QC standards that were carefully chosen not to interfere with the measurement of endogenous compounds wase spiked into every analyzed sample, allowed instrument performance monitoring, and aided chromatographic alignment. Instrument variability was determined by calculating the median relative standard deviation (RSD) for the standards that were added to each sample prior to injection into the mass spectrometers. Overall process variability was determined by calculating the median RSD for all endogenous metabolites (i.e., non-instrument standards) present in 100% of the pooled matrix samples. Experimental samples were randomized across the platform run with QC samples spaced evenly among the injections.

*UPLC-MS/MS:* All methods used a Waters ACQUITY UPLC and a Thermo Scientific Q-Exactive high resolution/accurate mass spectrometer interfaced with a heated electrospray ionization (HESI-II) source and Orbitrap mass analyzer operated at 35,000 mass resolution. The sample extract was dried then reconstituted in solvents compatible to each of the four methods. Each reconstitution solvent contained a series of standards at fixed concentrations to ensure injection and chromatographic consistency. One aliquot was analyzed using acidic positive ion conditions, chromatographically optimized for more hydrophilic compounds. In this method, the extract was gradient eluted from a C18 column (Waters UPLC BEH C18−2.1 × 100 mm, 1.7 μm) using water and methanol, containing 0.05% perfluoropentanoic acid (PFPA) and 0.1% formic acid. Another aliquot was also analyzed using acidic positive ion conditions; however, it was chromatographically optimized for more hydrophobic compounds. In this method, the extract was gradient eluted from the same afore mentioned C18 column using methanol, acetonitrile, water, 0.05% PFPA, and 0.01% formic acid and was operated at an overall higher organic content. Another aliquot was analyzed using basic negative ion optimized conditions using a separate dedicated C18 column. The basic extracts were gradient eluted from the column using methanol and water, but with 6.5 mM ammonium bicarbonate at pH 8. The fourth aliquot was analyzed via negative ionization following elution from a hydrophilic interaction liquid chromatography column (Waters UPLC BEH Amide 2.1 × 150 mm, 1.7 μm) using

a gradient consisting of water and acetonitrile with 10 mM ammonium formate, pH 10.8. The mass spectrometry (MS) analysis alternated between MS and data-dependent MS[n] scans using dynamic exclusion. The scan range varied slighted between methods but covered 70–1000 $m/z$. Raw data files were archived and extracted as described below.

*Bioinformatics:* The informatics system consisted of four major components: the Laboratory Information Management System (LIMS); the data extraction and peak identification software; data processing tools for QC and compound identification; and a collection of information interpretation and visualization tools for use by data analysts. The hardware and software foundations for these informatics components were the LAN backbone and a database server running Oracle 10.2.0.1 Enterprise Edition.

*Data extraction and compound identification:* Raw data were extracted, peak-identified, and QC processed using Metabolon's hardware and software. These systems are built on a web-service platform utilizing Microsoft's.NET technologies, which run on high-performance application servers and fiber-channel storage arrays in clusters to provide active failover and load-balancing. Compounds were identified by comparison to library entries of purified standards or recurrent unknown entities. Metabolon maintains a library based on authenticated standards that contains the retention time/index (RI), mass to charge ratio ($m/z$), and chromatographic data (including MS/MS spectral data) on all molecules present in the library. Furthermore, biochemical identifications are based on three criteria: retention index within a narrow RI window of the proposed identification, accurate mass match to the library ±10 ppm, and the MS/MS forward and reverse scores between the experimental data and authentic standards. The MS/MS scores are based on a comparison of the ions present in the experimental spectrum to the ions present in the library spectrum. While there may be similarities between these molecules based on one of these factors, the use of all three data points can be used to distinguish and differentiate biochemicals. More than 3300 commercially available purified standard compounds have been acquired and registered into LIMS for analysis on all platforms for determination of their analytical characteristics. Additional mass spectral entries have been created for structurally unnamed biochemicals, which have been identified by virtue of their recurrent nature (both chromatographic and mass spectral). These compounds have the potential to be identified by future acquisition of a matching purified standard or by classical structural analysis.

*Curation:* A variety of curation procedures were carried out to ensure that a high quality data set was made available for statistical analysis and data interpretation. The QC and curation processes were designed to ensure accurate and consistent identification of true chemical entities, and to remove those representing system artifacts, mis-assignments, and background noise. Metabolon data analysts use proprietary visualization and interpretation software to confirm the consistency of peak identification among the various samples. Library matches for each compound were checked for each sample and corrected if necessary.

*Metabolite quantification and data normalization:* Peaks were quantified using area under the curve. For studies spanning multiple days, a data normalization step was performed to correct variation resulting from instrument inter-day tuning differences. Essentially, each compound was corrected in run-day blocks by registering the medians to equal one (1.00) and normalizing each data point proportionately. For studies that did not require more than one day of analysis, no normalization was necessary other than for purposes of data visualization. In certain instances, biochemical data may have been normalized to an additional factor (e.g. cell counts, total protein as determined by Bradford assay, osmolality, etc.) to account for differences in metabolite levels due to differences in the amount of material present in each sample. The analysis yielded a dataset comprising compounds of known identity (referred to as biochemicals) with 557 named biochemicals in serum and 553 named biochemicals in feces. Metabolic pathways were visualized using the Cytoscape plugin in the Metabolync Portal (https://portal.metabolon.com). We used XLSTAT for Microsoft Excel (Addinsoft SARL, Paris, France)[37] and SIMCA 14.1 (Umetrics, Stockholm, Sweden)[64] softwares to build the PCA and the partial least squares-discriminant analysis (PLS-DA) models. The primary advantage of PCA and PLS-DA models is that the leading sources of variability in the data are modeled by new variables that explain most of the variance in the data and, consequently, in their associated scores and loadings, allowing the visualization and understanding of different patterns and relations in the data. PCA is able to find low-dimensional embedding of multivariate data in a manner that optimally preserves the structure of the data. PCA transforms variables in a data set into a smaller number of new latent variables called principal components (PCs), which are uncorrelated to each other and which account for decreasing proportions of the total variance of the original variables. Each new PC is a linear combination of the original variation such that a compact description of the variation within the data set is generated. Observations are assigned scores according to the variation measured by the PC with those having similar scores being clustered together.

PLS-DA is a classification technique that encompasses the properties of partial leastsquares regression with the power of discriminant analysis[67]. From a mathematical point of view, PLS-DA is a supervised extension of PCA used to distinguish two or more classes by searching for variables ($X$ matrix) that are correlated to class membership ($Y$ matrix). In this approach, the axes are calculated to maximize class separation and can be used to examine separation that would otherwise be across three or more principal components. PLS-DA model quality

was validated using $Q^2$ cum, $R^2Y$ cum and $R^2X$ cum values and CV-ANOVA was used to test the significance of the models ($P < 0.05$). Four groups and pairwise comparisons were clearly discriminated the groups from each other by the primary component $t$ (1) or the secondary component $t$ (2) based on the model quality parameters we used[67]. In addition, the PLS-DA models were validated by a permutation test. $R$ intercept and $Q$ intercept values were checked to see whether the models were not over fitted[67]. The "variable importance in project" (VIP) plots were generated to identify metabolites contributing significantly to the separation of the four genotypes. A cutoff value of 0.7–0.8 for the VIP is generally acceptable. In this study, the cutoff value was set at 1.0 (ref. [67]).

**Measurement of LPS concentration.** Serum LPS concentrations were measured with a ToxinSensor Chromogenic Limulus Amebocyte Lysate (LAL) Endotoxin Assay Kit (GenScript), following the manufacturer's instructions[37]. Briefly, to minimize inhibition or enhancement by contaminating proteins, the samples ($n = 9$–10 per group) were diluted 10- to 50-fold with endotoxin-free water, adjusted to the recommended pH, and heated for 10 min at 70°C. To obtain an endotoxin stock solution, the lyophilized endotoxin standard was dissolved by adding 2 ml of LAL reagent water and mixed thoroughly for 15 min with a vortexer. LAL reagents were added to serum and incubated at 37°C for 45 min, and the absorbance was read at 545 nm. A spiked control at 0.45 EU per ml was included for each sample to check that no significant inhibition or activation occurred. The lower limit of detection (LLOD) was 0.01 EU per ml. The coefficient of variation equals 100 times the standard deviation of a group of values divided by the mean and is expressed as a percent. The coefficient of variation absorbance was less than 10%.

**Measurement of cytokine levels and other circulating factors.** Serum samples ($n = 9$–10 per group) were analyzed for levels of TNF-α (LLOD:1.4 pg ml$^{-1}$), IL-1β (LLOD: 9.4 pg ml$^{-1}$), IL-6 (LLOD: 0.2 pg ml$^{-1}$) and MCP-1 (LLOD: 3.7 pg ml$^{-1}$) by Bio-Plex immunoassays (assay range: 2–3000 pg ml$^{-1}$; intra-assay coefficient of variation: <10%; inter-assay coefficient of variation: <30%) formatted on magnetic beads (Bio-Rad Laboratories Inc, CA, USA), following the manufacturer's instructions[37]. Xponent software (Luminex, Austin, TX) was used for data acquisition and analysis. ELISA kits were used to analyze serum levels of sCD14 (LLOD: 0.06 ng ml$^{-1}$; inter- and intra-assay coefficients of variation were <12 and <8%, respectively) (MyBioSource, San Diego, CA) and LBP (LLOD: 0.4 ng ml$^{-1}$; coefficient of variation %: <6) (NeoBioLab, Cambridge, MA), according to the manufacturers' instructions. For all the assays mentioned previously, 5–6 standards including blank (negative control) were used.

**Determination of fatty acid composition of mouse tissues and diets.** Fatty acid profiles of mouse diets and tail tissues ($n = 9$–10 per group) were analyzed by GC as described previously[16,55]. Briefly, tissue or food samples were ground to powder under liquid nitrogen and subjected to total lipid extraction and fatty acid methylation by 14% boron trifluoride (BF3)-methanol reagent (Sigma-Aldrich) at 100°C for 1 h. Fatty acid methyl esters were analyzed using a fully automated HP5890 GC system equipped with a flame-ionization detector (Agilent Technologies, Palo Alto, CA). The fatty acid peaks were identified by comparing their relative retention times with the commercial mixed standards (NuChek Prep, Elysian, MN), and area percentage for all resolved peaks was analyzed by using a Perkin-Elmer M1 integrator.

**Measurement of intestinal permeability.** An intestinal permeability assay was performed as previously described[37]. Briefly, 6 h fasted mice ($n = 9$–10 per group) were gavaged with a phosphate buffer saline (PBS, pH 7.2) containing fluorescein isothiocyanate (FITC)-dextran (4 kDa; Sigma-Aldrich, St. Louis, MO) at a dose of 600 mg kg$^{-1}$ body weight. Blood samples (120 μl) were collected after 90 min from the facial vein. Serum was diluted with an equal volume of PBS, and fluorescence intensity was measured using a fluorospectrophotometer (Perkin-Elmer) with an excitation wavelength of 480 nm and an emission wavelength of 520 nm. Serum FITC-dextran concentration was calculated from a standard curve generated by serial dilution of FITC-dextran in PBS.

**Immunofluorescent staining.** Formalin fixed ileum, liver and epididymal white adipose tissue samples ($n = 3$ per group) and primary antibodies for ZO-1 (1:100; GTX108592) and TLR4 (1:100; GTX125909) from GeneTex (San Antonio, TX, USA) were given to experienced research technicians at the Massachusetts General Hospital (MGH) Core, Boston, MA. Briefly, deparaffinize slides, citrate antigen retrieval solution in pressure cooker, dual endogenous enzyme Block (5 min), normal goat serum (20 min), incubation with primary antibody overnight at 4 C and then rabbit polymer HRP (30 min) followed by DAB+ (5–10 min). All samples were photographed using an immunofluorescence microscope (LSM710; Zeiss) and analyzed for ZO-1 and TLR4 expression with the help of a pathologist fellow at MGH Core[37].

**Liver histopathology and hepatic steatosis scoring.** Frozen liver samples were stained with Oil Red O and Masson's trichrome (MGH Core, Boston, MA) and subcutaneous white adipose and inter-scapular brown adipose tissues were stained with hematoxylin & eosin (MGH Core, Boston, MA). Oil Red O-stained liver slides were examined by an independent experienced pathologist blinded to group assignment. Hepatic steatosis was graded based on the number and size of stained fat droplets[37]: 0 (none); 1+ (<5%); 2+ (6–33%); 3+ (34–66%); 4+ (>66%); 5+ (>66% plus very large fat globules).

**Glucose tolerance test.** Glucose tolerance test (GTT) was performed in non-anesthetized mice as described[37]. Briefly, mice ($n = 9$–10 per group) were fasted overnight, fasting blood sugar was measured, and glucose [1.0 g kg$^{-1}$ body weight, 20% (wt/vol) glucose solution] was administered by oral gavage. Blood samples (microliters) were drawn from the tip of the incised tail at 15, 30, 60, 90, and 120 min to measure blood glucose levels. Glucose tolerance was assessed by calculating the incremental area under the curve of each GTT.

**Statistics and reproducibility.** Data were expressed as mean ± standard error of mean (SEM). Box-plots (box showing the median, and the 25th and 75th percentiles, and the whiskers of the graph show the largest and smallest values) were also used to express the data. Multivariate analyses (PCA/PCoA/Taxon significance testing/PLS-DA/PERMANOVA/non-parametric differential expression) of omics data were performed using the Secondgenome R package: 0.2.4, DESeq2 package[59], PAST version 3.11 (ref. [37]), XLSTAT (2017.6) for MS Excel (Addinsoft SARL, Paris, France)[37] and SIMCA 14.1 (Umetrics, Stockholm, Sweden)[64] as explained in the Methods section. Gephi Graph Visualization and Manipulation software version 0.9.2 (ref. [37]) was used to visualize the network. ImageJ software was utilized to draw scale bars on histopathological pictures. Univariate analyses (t-tests, ANOVA and correlation analysis) were performed using Prism 8.0 (GraphPad Software, Inc.) and PAST version 3.11 and SIMCA 14.1. Heat maps were generated using XLSTAT software version 2017.6 and Prism 8.0 (GraphPad Software, Inc.). Statistical differences among four groups in other data were evaluated by ordinary one-way analysis of variance (ANOVA) with Tukey's or non-parametric Kruskal–Wallis test with Dunn's multiple comparison post-tests (GraphPad Prism 8). Data were checked for heterogeneous variance with the Bartlett's and Brown-Forsythe tests or the $F$ test. If unequal variance was detected, data were analyzed using non-parametric tests. If parametric and non-parametric analyses did not show statistical differences, actual data were used for analysis without transformation and parametric analysis results were presented. The significance was considered to be at $P < 0.05$.

**Reporting Summary.** Further information on research design is available in the Nature Research Reporting Summary linked to this article.

## Data availability

Source data of figures, OTU tables, raw data, taxonomy, FASTA files, KEGG and KO pathways and abundance data, metadata for 16S rRNA gene sequence analysis as well as metabolomics data generated in this study have been made publicly available in Figshare[68] (https://figshare.com/s/89bd8826c65b96c7e197; https://doi.org/10.6084/m9.figshare.7607441). Figures that have associated raw data are Figs. 1–6, and Supplementary Figs. 1, 4 and 5.

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

## Acknowledgements

This study was supported by the generous funding from Sansun Life Sciences and the Fortune Education Foundation. The authors are also grateful to Marina Kang for her editorial assistance.

## Author contributions

J.X.K. and K.K. conceived and designed the study; K.K. collected serum, fecal, and tissue samples for metagenomics and metabolomics and immunohistochemical analyses, performed qPCR-based bacterial quantification, measured the markers of metabolic endotoxemia and chronic low-grade inflammation, analyzed all the data and performed univariate and multivariate statistical analyses including network construction and PLS models for inter-omic and host–microbiome interaction analyses; X.-Y.L. and B.W. prepared the transgenic mouse models (animal breeding, genotyping and phenotyping) and measured the markers of metabolic syndrome (body weight gain, body composition analysis, fatty liver, and glucose tolerance test) and liver cancer. Q.P. performed brown and white adipose tissue histopathological analyses, body composition analysis, and energy metabolism experiments; C.-Y.C., L.H. and S.X., contributed to fatty acid analysis and samples collection; K.K. and J.X.K. wrote the manuscript; All authors approved the final version of the manuscript.

## Additional information

**Competing interests:** The authors declare no competing interests.

