## [Peer Review File · Communications Biology]

Reviewers' comments:

Reviewer #1 (Remarks to the Author):

The article entitled « Multi-Omic Analysis of Transgenic Animal Models Uncovers the Tissue Omega-6/Omega-3 Fatty Acid Imbalance as a Critical Risk Factor for Chronic Disease » by Kanakaraju Kaliannan et al., is well written, the experiments carried out and the results partially support the conclusions of the article, and brings new arguments about the importance of omega-3/omega-6 intake in maintaining a state of equilibrium. However, before publication, authors must address the following points:

- 1- There are no precise definition of "balanced" and "unbalanced" ratio? From which ratio is it considered unbalanced by the authors? How much is the ratio in fine? Is it physiological and transposable in humans?
- 2- The animal model used is very relevant and avoids modulating the nutritional intake to change the ratio W3/W6. However, the "health" effects observed on the various parameters mentioned do not take into account the lifestyle (in particular the level of physical activity). This is an important bias to translate these results. Were the mice in a collective or individual cage, what environment (ie, physical activity level)? It is a limit to integrate in discussion section. Mastering nutrition without mastering the energy expenditure can not make it possible to judge the expected effects on health (lipid profile, glycemic balance, inflammation, SO, microbiota ...). Authors should discuss this point.
- 3- Did the authors calculate and/or analyze the ratios on their different groups of transgenic mice used in this study? Is the FAT 1 + 2 ratio physiologically transferable to humans? There are no data/figures on it.
- 4- In the results section, the authors do not present numerical data and no statistical analysis when possible. In addition, it would be more logical not to lose the reader to present the results in the following order: first body composition then the markers of inflammation and metabolic disorders, and finish with the intestinal microbiota data and metabolomic results.
- 5- In the figures, there are many graphics and data presented on the same page, the figures are illegible. To improve the presentation, it would be important to summarize the important data as table. Too much data kills the main message.
- 6- Figure 6C: Diagram is build upside down? Authors said « Diagram illustrating our proposed mechanism of tissue omega-6 / omega-3 fatty acid imbalance leading to the development of chronic disease ». So the imbalance is due to the increase of W3 and the decrease of W6? Is it not the opposite? Authors could explain ?

Reviewer #2 (Remarks to the Author):

Summary. The health-related importance of balance in dietary and tissue levels of omega-6 (n-6) and omega-3 (n-3) polyunsaturated fatty acids (PUFAs) has long been a subject of debate. A confounding factor for research in this area is that n-6 and n-3 PUFAs are essential in the mammalian diet and controlling for amounts of each in feeding studies is challenging and/or unappreciated. The manuscript authors have a unique set of transgenic mice that can be used to specifically alter n-6 and/or n-3 content in the mouse tissues, without having to alter the mouse chow between test groups. This manuscript describes an in-depth metabolic phenotyping, multi-omics analysis of tissues, serum, and fecal samples from wild type mice, mice producing, in comparison to wild type, excess n-6, excess n-3, and excess n-6 and n-3 PUFAs. The authors find significant gut microbiota, metabolomic, and metagenomic patterns that distinguish high n-6 levels (and n-6/n-3 ratio) from lower n-6 levels and balanced (1:1) ratio of N-6:n-3, specifically patterns reflecting dysbiosis, pro-inflammatory state, and

disease-associated phenotypes and biomarkers.

Overall impression. Overall, I think that this is a very important study of unique design that cannot be replicated with other genetic mouse models. The findings support prior evidence that dietary PUFAs are highly influential in health mediating effects, with n-6 being pro-inflammatory. Further, these studies uniquely show effects of tissue source n-6 and n-3 PUFAs on the gut microbiota and networks of coordination between microbiota, circulating inflammatory factors and metabolites, metabolic phenotypes and cancer risk. This manuscript has key implications for dietary guideline discussions and future human dietary intervention studies.

Specific comments.

1. Unnecessary use of non-standard abbreviations. CLGI is not a standard abbreviation. Should just spell out "Chronic low-grade inflammation"; same for ME metabolic endotoxemia. Confusing to most readers. HCN hierarchical clustering (only used 2x) should be spelled out for both usages.
2. Need to define what "balanced tissue ratio of n-6/n-3 PUFA" means in abstract and introduction. Is this equal 1:1 ratio? Other?
3. There seems to be a mis-statement that needs revision in the paragraph preceding the Results section:
4. "Consequently, we have four genotypes of mice for use: wild-type (WT) (incapable of producing essential fatty acids), FAT-1 (producing n-3 fatty acids), FAT-2 (producing only n-6 fatty acids), and FAT-1+2 (producing both n-6 and n-3 fatty acids) [15]. These mice exhibit four distinct PUFA phenotypes varying in the quantity of PUFA and n-6/n-3 ratio, even though they are all fed an identical diet with no need of dietary PUFA supplementation [15]." This last statement is false: WT mice need dietary n-6 and n-3 PUFA, FAT-1 mice need dietary n-6 PUFA, FAT-2 mice need dietary n-3 PUFA. Only FAT-1+2 mice can survive without any n-3 or n-6 PUFA supplementation.
5. Page 5, line 7. There is no such thing as "metabolomics sequencing."
6. Page 5. P-value shown should not have a comma.
7. Page 6. The approach described starting with the last sentence seems quite biased. I think that the authors should use an unbiased statistical approach to determine the pathways that are differentially expressed in the metagenomics data between the groups, rather than fishing in pathways that they are interested in.
8. Authors should note in Discussion that findings may not be translational to females, as only male mice were studied.
9. The multi-/inter-omics results sections are overly lengthy and should be abbreviated significantly. The detail is too great for this journal and readership, in my opinion.
10. Page 9, bottom paragraph. Where statistical analyses run on the GTT data? The text states that "glucose tolerance testing (GTT) showed that FAT-1 and FAT-1+2 mice exhibited better glucose tolerance than the WT and FAT-2 mice." I see no evidence of any statistical support of this statement.
11. The body weight of the FAT-2 mice is more than 10% greater than WT. Yet, here is not difference between these 2 groups in fat or lean mass or fluid weight. To what do the authors attribute this body weight difference? Fig 3 I, J, K
12. Was food intake measured during the study at all? Energy expenditure recorded? FAT-2 mice have heavier body weight than FAT-1 and FAT-1+2 mice. This could be due to altered food intake and/or energy expenditure. Should be noted in text and discussion either way.

Figures

13. Authors must provide legends for the Supplementary figures.
14. Fig 1. Panels A (insert), C, D, H (insert), and I. Font is too small, illegible even at high magnification. What are the units for the scales shown in Panels C and I?
15. Fig 1 & 2. Figure titles should be changed for greater accuracy. The ratio does not affect the changes, it is the tissue-generated alteration in n-6 and n-3 PUFA production (and thus ratio) that affects the changes. Suggest, "Alterations in tissue-generated n-6/n-3 ratio impact..."
16. Fig 3 Panel q. The photos need labels. No way to know what mouse groups these come from.

17. Fig S1. Panel K. Font is too small, illegible even at high magnification. Order of the groups should match the order in the other panels, especially when the font is nearly impossible to read. Don't know what this data is showing as there is no legend and it is not described in the text.

18. Fig S2. Panels A, C, and I. Font is too small, illegible even at high magnification.

Supplementary Data (no page numbers provided)

19. Animal Diets section: Typographical error(?) - "fat-12" or "Fat12" mice. This nomenclature is inconsistent with the body of the manuscript (where the strain is referred to as FAT-1+2) and is misleading. Looks like "twelve". Occurs here and other places in the supplementary data document.

20. The complete fatty acid composition details of the diet should be provided, including fatty acid specifics (e.g., amount of linoleic acid, alpha linolenic acid, arachidonic acid, palmitate, etc.) as well as SFA, PUFA and MUFA proportions.

21. Coefficients of variance for the measurement of cytokines and other circulating factors as well as LLOD should be reported for each assay. A description of the assay controls used should also be described.

22. Methods here describe insulin tolerance tests (ITT), insulin measurements, and HOMA-IR calculations. None of the data is presented. This data must be presented if the studies were conducted.

Dear Editor,

Thank you and the reviewers for the careful review and helpful comments. We have addressed thoroughly the reviewers' concerns point by point as follows and made changes to the manuscript accordingly (with tracking mark). We trust that the manuscript has now been significantly improved and should be acceptable for publication.

We thank you again for your reconsideration of the revised manuscript and look forward to hearing from you soon.

Sincerely,

Dr. Kanakaraju Kaliannan (Lead author) and Dr. Jing X Kang (Senior Author)

Reviewers' comments:

Reviewer #1:

The article entitled « Multi-Omic Analysis of Transgenic Animal Models Uncovers the Tissue Omega-6/Omega-3 Fatty Acid Imbalance as a Critical Risk Factor for Chronic Disease » by Kanakaraju Kaliannan et al., is well written, the experiments carried out and the results partially support the conclusions of the article, and brings new arguments about the importance of omega-3/omega-6 intake in maintaining a state of equilibrium. However, before publication, authors must address the following points:

1- There are no precise definition of "balanced" and "unbalanced" ratio? From which ratio is it considered unbalanced by the authors? How much is the ratio in fine? Is it physiological and transposable in humans?

The Western human diet today is fundamentally different from that throughout the majority of human evolution (PMID: 2981409, PMID: 3578100). Evolutionarily, the omega-6/omega-3 fatty acid ratio would have been closer to 1: 1. Modern humans consume much more omega-6 fatty acids and less omega-3 fatty acids than their ancestors and have an elevated omega-6 to omega-3 fatty acid ratio ranging from 10: 1 to 50: 1, due to the industrialization of agriculture, processed foods, grain-fed livestock, and the increased consumption of vegetable oils (PMID: 21367944). This shift in the omega-6/omega-3 ratio has profound implications on human health (PMID: 21279554). For example, a ratio of 2-3/1 suppressed inflammation in patients with rheumatoid arthritis, and a ratio of 5/1 had a beneficial effect on patients with asthma, whereas a ratio of 10/1 had adverse consequences (PMID: 18408140). Subjects with a low dietary n-6: n-3 PUFA ratio (first tertile) had better triglyceride, VLDL-c, glucose, insulin, and HOMA-IR parameters than those with a higher dietary ratio (third tertile) (PMID: 30092569). A ratio of 2.5/1 reduced rectal cell proliferation in patients with colorectal cancer, whereas a ratio of 4/1 with the same amount of omega-3 PUFA had no effect (PMID: 18408140). The lower omega-6/omega-3 ratio in women with breast cancer was associated with decreased risk. An interventional study showed that administration of high concentrate omega-3 for 24 weeks significantly decreased the

RBC omega-6/omega-3 fatty acid ratio, which was associated with a significant reduction in liver fat content in non-alcoholic fatty liver disease (PMID: 30127297). These studies indicate that an optimal or beneficial ratio may vary with the disease and individual conditions and suggest that reduction of a relatively higher tissue ratio to a lower one is beneficial. Our study reported in this manuscript, for the first time by using the unique animal models plus multi-omics, provides clear and compelling evidence for such a relationship between the ratio and health status.

Although there is no precise definition of "balanced" and "unbalanced" ratio, it is generally believed that a balance between omega-6 and omega-3 fatty acids (a ratio of around 1:1) is ideal for good health. "Balance" refers to a ratio close to 1:1, whereas "Balanced" means that a high ratio has been reduced toward 1:1 (but does not have to be 1:1, just relatively lower than a control level, for example, reduction from 5:1 to 2:1). So, "a balanced ratio" can also be denoted as "a reduced or lower ratio."

2- The animal model used is very relevant and avoids modulating the nutritional intake to change the ratio W3/W6. However, the "health" effects observed on the various parameters mentioned do not take into account the lifestyle (in particular the level of physical activity). This is an important bias to translate these results. Were the mice in a collective or individual cage, what environment (i.e., physical activity level)? It is a limit to integrate in discussion section. Mastering nutrition without mastering the energy expenditure can not make it possible to judge the expected effects on health (lipid profile, glycemic balance, inflammation, SO, microbiota ...). Authors should discuss this point.

First of all, we thank the reviewers for the positive comments on the transgenic mice model used in this study.

We totally agree that mastering nutrition without mastering the energy expenditure cannot make it possible to judge the expected effects on health. The results of energy metabolism experiments have been mentioned in the results section of the revised manuscript as follows:

“The body weight of adult mice at the age of 8 months differed significantly between the genotypes, with the highest body weights for the FAT-2 group and the lowest body weights for the FAT-1 and FAT-1+2 groups (**Fig. 1a**) although no significant differences in the food intake (**Supplementary Fig. 2a**). Usually, the difference in fat mass observed during the young age would diminish when they become older. We thus followed up the phenotype of these mice for a prolonged WD exposure (16 months). Even after 16 months, FAT-2 group maintained the highest body weight and FAT-1+2 maintained the lowest body weight (**Supplementary Fig. 2b**). Increased body weight was correlated with increases in body fat mass, especially abdominal fat (**Fig. 1b-c**) in the 16-month-old mice. To assess the cause of this persisting body weight difference between groups, especially between FAT-2 and FAT-1+2, we investigated energy metabolism at the age of 16 months. At that point, there were no differences in the food intake between groups (**Supplementary Fig. 2c**). However, FAT-2 mice spent significantly less energy than FAT-1+2 and FAT-1 mice in the dark and light phases (**Supplementary Fig. 2d-e**), together with lower CO₂ production (**Supplementary Fig. 2f-g**), O₂ consumption (**Supplementary Fig. 2h-i**). The respiratory exchange ratio (RER) was unaltered

(**Supplementary Fig. 2j-k**). Moreover, it is notable that this difference in the energy expenditure was not due to a difference in physical activity (**Supplementary Fig. 2l-m**)”.

We have also discussed the energy metabolism related details in the discussion section of the revised manuscript as follows:

“A recent study has shown that male FAT-1 mice fed high-fat diet for 6 weeks at young age showed lean phenotype associated with higher energy expenditure than WT counterparts³⁰. Conversely, the sustained increase in fat mass and eventually body weight observed in FAT-2 mice in this study was likely due to another mechanism, which is the lower energy expenditure (EE). The altered energy homeostasis (lower EE, CO₂ production and O₂ consumption) in the FAT-2 mice with elevated tissue n-6/n-3 ratio could partially be due to impaired browning process³¹ and altered intestinal endocannabinoid system (ECS)³¹. However, this assumption warrants further investigation”.

Please kindly note that the n values in the originally submitted manuscript were 6-7 for all the data related to body weight (Supplementary Fig. 3i), visceral white adipose tissue weight (Supplementary Fig. 3j) body composition (Fig. 3k), glucose tolerance test with area under the curve (Fig. 3l), liver cancer (Fig. 3q and 3r), tail tissue fatty acid analysis (Fig. 1a-j), relative abundance of family *Enterobacteriaceae* and genus *Bifidobacterium* (Fig. 2d and 2e) and markers of metabolic endotoxemia (Fig. 3a-d) and chronic low-grade inflammation (Fig. 3e-h), in order to match with those used for the omics analysis. Based on the reviewers advise and suggestions, we now have included all animals used and re-analyzed these data using new n values [(WT) (n=9), FAT-2 (n=10), FAT-1 (n=10) and FAT-1+2 (n=10)] and presented the revised results and figures accordingly. Please refer new Fig. 1, Fig. 1(a-k), Fig. 2 (including the energy metabolism data) and Fig. 4d-e in the revised manuscript. The mice were housed in 2 or 3 per cage during the study.

3- Did the authors calculate and/or analyze the ratios on their different groups of transgenic mice used in this study? Is the FAT 1 + 2 ratio physiologically transferable to humans? There are no data/Figures on it.

Yes. The PUFA phenotypes of the four groups of mice used in this study (WT, FAT-2, FAT-1 and FAT-1+2 mice) has been reported previously (reference 15 in the revised manuscript). The tissue n-6 and n-3 PUFA content in these mice was further validated and shown in supplementary Fig. 1. We analyzed the n-6/n-3 PUFA ratios in the tail and serum samples of WT, FAT-2, FAT-1 and FAT-1+2 groups. Please refer supplementary figures 1d and 1p. The n-6/n-3 PUFA ratios in the tissues of FAT-1+2 mice were approximately 1-2:1, which is well physiologically transferable to humans because several sources of information suggest that human beings evolved on a diet with a ratio of n-6 to n-3 essential fatty acids of approximately 1 (PMID: 17045449). Many healthy populations today, like Japanese and Mediterranean people, have a tissue n-6/n-3 ratio of close to 1-2:1 (PMID: 16387724, PMID: 12955797, PMID: 16841858). As another example for human tissue ratio, adipose tissue n-6/n-3 ratios in control subjects and breast cancer cases from Spain country were 1: 1.13 (table 4 of PMID: 29422481)

and 3:1 (table 1 of PMID: 9508101), respectively. Likewise, n-6/n-3 ratios in red blood cell membranes of breast cancer cases and control women in Shanghai, China were 3.7:1 and 3.6: 1 (table 3 of PMID: 17413110), respectively.

4- In the results section, the authors do not present numerical data and no statistical analysis when possible. In addition, it would be more logical not to lose the reader to present the results in the following order: first body composition then the markers of inflammation and metabolic disorders, and finish with the intestinal microbiota data and metabolomic results.

Thanks for the suggestion and we agree with the reviewer. Although we wanted to present the numerical data and statistical analysis in the results section, we had to follow the word limit (5000 words for main text) required by the journal. We have added the numbers when possible. We have also followed the reviewer's suggestion to rearrange the order for the Figures: Fig. 3 became Fig. 1 (body composition, the markers of inflammation and metabolic disorders), Fig. 1 and 2 became Fig. 2 (gut microbiota) and 3 (fecal and serum metabolites) and related results in the revised manuscript.

When we rearranged the supplementary Figures according to the new order for main Figures, we found that original **Supplementary Fig. 4j** in the initially submitted manuscript should have shown the correlation between fecal indolepropionate (IPA) and intestinal permeability (IP) instead of showing the correlation between fecal and serum IPA, which had already been shown in the **Supplementary Fig. 4e**. In the revised manuscript, we have added a new Fig. (**Supplementary Fig. 5k**) showing the correlation between fecal IPA and IP. We sincerely apologize for this accidental mistake related to Fig. although we correctly mentioned the results (the correlation between IPA and IP) in the original submission.

In addition, in the supplementary Figures, we have added two new Figures (**Supplementary Fig. 5l-m**) showing correlations between fecal levels of n-6 PUFA (linoleic acid and arachidonic acid) and visceral adiposity. A recent 'nature genetics' journal article (PMID: 29808030) has shown a significant and positive correlation between fecal arachidonic acid and visceral adiposity and between fecal arachidonic acid and body mass index. Interestingly, we found similar associations with our data, so we have included these important correlations (**Supplementary Fig. 5l-m** and supplementary Table 2 including arachidonic acid and related reference) in the revised manuscript.

Also, we have deleted the 'n3', which was duplicated in the supplementary Table 2 (Docosapentaenoate (n3DPA; 22:5n3).

5- In the Figures, there are many graphics and data presented on the same page, the Figures are illegible. To improve the presentation, it would be important to summarize the important data as table. Too much data kills the main message.

To improve the legibility of Figures, we have modified the Figures (aligned dot plots showing individual values) with increased the font size and/or panel size in the Figures.

While we agree that it would be important to summarize the important data as table, we still think that it is necessary to keep the Figures to show the relationships of the tissue n-6/n-3 PUFA ratio with gut microbiota composition (**Supplementary Fig. 2c**) and function (**Supplementary Fig. 2i**), fecal (**Fig. 3b**) and serum (**Fig. 3p & 3q**) metabolites, as well as their associations with metabolic disorders, gut barrier integrity, inflammation and cancer. Some related information and references have been presented in supplementary Table 1, Table 2 and Table 3. The key points have been mentioned in the Results section. Overall, our data presented in this manuscript collectively show a positive association of a high tissue n-6/n-3 ratio (as in WT & FAT-2 mice) and a negative association of a balanced tissue n-6/n-3 ratio (as in FAT-1 & FAT-1+2 mice) with the development of dysbiosis, inflammation and chronic diseases (metabolic syndrome and cancer), which is the main message of this study.

6- Fig. 6C: Diagram is build upside down? Authors said « Diagram illustrating our proposed mechanism of tissue omega-6 / omega-3 fatty acid imbalance leading to the development of chronic disease ». So the imbalance is due to the increase of W3 and the decrease of W6? Is it not the opposite? Authors could explain ?

Thank you for bringing this to our attention. To match the legend with the details of the diagram, we have modified the Fig. 6c as in the revised manuscript.

Reviewer #2 (Remarks to the Author):

Summary. The health-related importance of balance in dietary and tissue levels of omega-6 (n-6) and omega-3 (n-3) polyunsaturated fatty acids (PUFAs) has long been a subject of debate. A confounding factor for research in this area is that n-6 and n-3 PUFAs are essential in the mammalian diet and controlling for amounts of each in feeding studies is challenging and/or unappreciated. The manuscript authors have a unique set of transgenic mice that can be used to specifically alter n-6 and/or n-3 content in the mouse tissues, without having to alter the mouse chow between test groups. This manuscript describes an in-depth metabolic phenotyping, multi-omics analysis of tissues, serum, and fecal samples from wild type mice, mice producing, in comparison to wild type, excess n-6, excess n-3, and excess n-6 and n-3 PUFAs. The authors find significant gut microbiota, metabolomic, and metagenomic patterns that distinguish high n-6 levels (and n-6/n-3 ratio) from lower n-6 levels and balanced (1:1) ratio of N-6:n-3, specifically patterns reflecting dysbiosis, pro-inflammatory state, and disease-associated phenotypes and biomarkers.

Overall impression. Overall, I think that this is a very important study of unique design that cannot be replicated with other genetic mouse models. The findings support prior evidence that

dietary PUFAs are highly influential in health mediating effects, with n-6 being pro-inflammatory. Further, these studies uniquely show effects of tissue source n-6 and n-3 PUFAs on the gut microbiota and networks of coordination between microbiota, circulating inflammatory factors and metabolites, metabolic phenotypes and cancer risk. This manuscript has key implications for dietary guideline discussions and future human dietary intervention studies.

The authors sincerely thank the reviewer for the positive comments.

Specific comments.

1. *Unnecessary use of non-standard abbreviations. CLGI is not a standard abbreviation. Should just spell out “Chronic low-grade inflammation”; same for ME metabolic endotoxemia. Confusing to most readers. HCN hierarchical clustering (only used 2x) should be spelled out for both usages.*

To avoid confusion and unnecessary use of non-standard abbreviations, we have followed your advice and wrote the full form for CLGI, ME and HCN.

2. *Need to define what “balanced tissue ratio of n-6/n-3 PUFA” means in abstract and introduction. Is this equal 1:1 ratio? Other?*

While “Balance” refers to a ratio close to 1:1, “Balanced” means that a high ratio has been reduced toward 1:1 (but does not have to be 1:1, just relatively lower than a control level, for example, reduction from 5:1 to 2:1). So, “a balanced ratio” can also be denoted as “a reduced or lower ratio”.

3. *There seems to be a mis-statement that needs revision in the paragraph preceding the Results section:*

4. *“Consequently, we have four genotypes of mice for use: wild-type (WT) (incapable of producing essential fatty acids), FAT-1 (producing n-3 fatty acids), FAT-2 (producing only n-6 fatty acids), and FAT-1+2 (producing both n-6 and n-3 fatty acids) [15]. These mice exhibit four distinct PUFA phenotypes varying in the quantity of PUFA and n-6/n-3 ratio, even though they are all fed an identical diet with no need of dietary PUFA supplementation [15].” This last statement is false: WT mice need dietary n-6 and n-3 PUFA, FAT-1 mice need dietary n-6 PUFA, FAT-2 mice need dietary n-3 PUFA. Only FAT-1+2 mice can survive without any n-3 or n-6 PUFA supplementation.*

Response to comments 3 & 4:

Actually, what we tried to say is that each genotype of the mice used has tissue abundance in a particular PUFA without the need of supplementation with that PUFA to their diet. For example, fat-1 mice have high tissue levels of omega-3 PUFA with no need of omega-3 supplementation for them as they can produce omega-3 PUFA themselves; Similarly, fat-2 mice have high tissue levels of omega-6 PUFA with no need of omega-6 supplementation for them as they can endogenously synthesize omega-6 PUFA. For clarity, we have revised the sentence as follows:

“They are all fed an identical diet with no need of dietary supplementation with **corresponding** PUFA.”

5. Page 5, line 7. *There is no such thing as “metabolomics sequencing.”*

Thank you. We have modified the sentence as follows:, we performed high-throughput metagenomic sequencing and metabolomics analysis of fecal samples.

6. Page 5. *P-value shown should not have a comma.*

Thank you. We replaced the ‘comma’ with a ‘period’.

7. Page 6. *The approach described starting with the last sentence seems quite biased. I think that the authors should use an unbiased statistical approach to determine the pathways that are differentially expressed in the metagenomics data between the groups, rather than fishing in pathways that they are interested in.*

Although we mentioned that we ‘selected’ several KEGG pathways, we first performed the non-parametric ‘differential expression analysis’ (including Benjamini-Hochberg false discovery rate (FDR) corrected P value <0.05) on the relative abundance of microbial function related genes and KEGG pathways as we mentioned in the supplementary methods (please refer ‘microbiome functional analysis’). We then selected those KEGG pathways with a FDR corrected P value <0.05 to check whether they were associated with metabolic syndrome, inflammation, bacterial translocation, non-alcoholic fatty liver disease and cancer (Table 1) based on the recently published articles. To make it clear, we modified this sentence as follows:

Next, we selected a number of differentially expressed KEGG pathways (FDR corrected $P <0.05$) associated with.....

8. *Authors should note in Discussion that findings may not be translational to females, as only male mice were studied.*

Yes. As advised, we have mentioned it in the discussion section of revised manuscript as follows:

“The necessity of conducting animal research in the female mice because our findings obtained with male mice may not be translational to females”.

9. The multi-/inter-omics results sections are overly lengthy and should be abbreviated significantly. The detail is too great for this journal and readership, in my opinion.

We understand the reviewer’s concern that multi-/inter-omics results sections are overly lengthy. There are two reasons we would sincerely would like to say to keep this section as it is. 1) After reading several related high impact articles, we summarized the ‘inter-omics’ results to this level. When we further abbreviated it, we ourselves found it difficult to understand and we were afraid that readers at junior levels and/or those who are not working in the field might have difficulty to understand them. 2) In addition to transgenic mouse models used in this study, we believe that results obtained from inter-omics and host-microbiome interaction analyzes are very important data that enable us to identify the tissue omega-6/omega-3 fatty acid imbalance as a critical risk factor for chronic disease.

10. Page 9, bottom paragraph. Where statistical analyses run on the GTT data? The text states that “glucose tolerance testing (GTT) showed that FAT-1 and FAT-1+2 mice exhibited better glucose tolerance than the WT and FAT-2 mice.” I see no evidence of any statistical support of this statement.

Thank you for bringing this to our attention. We have added the statistical support. Please refer Fig. 1f (Fig. 3l in the original manuscript) and its legend (* FAT-2 vs. FAT-1 and FAT-2 vs. FAT-1+2. & WT vs. FAT-2. # WT vs. FAT-1) in the revised manuscript.

11. The body weight of the FAT-2 mice is more than 10% greater than WT. Yet, here is not difference between these 2 groups in fat or lean mass or fluid weight. To what do the authors attribute this body weight difference? Fig 3 I, J, K

Thank you very much for your careful review and the thoughtful question. Before answering your question, we would kindly like to let you the following changes in the revised manuscript. The n values in the originally submitted manuscript were 6-7 for all the data related to body weight (Supplementary Fig. 3i), visceral white adipose tissue weight (Supplementary Fig. 3j) body composition (Fig. 3k), glucose tolerance test with area under the curve (Fig. 3l), liver cancer (Fig. 3q and 3r), tail tissue fatty acid analysis (Fig. 1a-j), relative abundance of family *Enterobacteriaceae* and genus *Bifidobacterium* (Fig. 2d and 2e) and markers of metabolic endotoxemia (Fig. 3a-d) and chronic low-grade inflammation (Fig. 3e-h), in order to match with those used for the omics analysis. Based on the reviewers advise and suggestions, we now have included all animals used and re-analyzed these data using new n values [(WT) (n=9), FAT-2 (n=10), FAT-1 (n=10) and FAT-1+2 (n=10)] and presented the revised results and Figures accordingly. Please refer new Fig. 1, Fig. 1a-k, Fig. 2 (including the energy metabolism data) and Fig. 4d-e in the revised manuscript.

To answer your question, the body weight (Supplementary Fig. 3i in the original submission) of the FAT-2 mice was greater compared to WT when these mice were 8-month-old (the mouse age was mentioned in the results section of the original submission). However, the body composition data (Fig. 3j&3k in the original submission) was obtained when we conducted energy metabolism studies in these mice at the age of 16 months. The observed body weight differences between FAT-2 and WT at the age of 8 months might have been due to differences in the fat mass and/or lean mass and/or fluid weight, which we did not confirm because we did not analyze the body composition when these mice were 8-month-old. In the revised manuscript, we have shown the body weight (**Supplementary Fig. 2b**) taken at the age of 16 months. Accordingly, we observed only a small non-significant increase in the body weight of FAT-2 mice compared to WT mice. We believe that this body weight (Supplementary Fig. 2b in the revised manuscript) and body composition (**Supplementary Fig. 1c** in the revised manuscript) data measured at the same time at the age of 16 months could answer to the reviewer's question in this regard.

12. Was food intake measured during the study at all? Energy expenditure recorded? FAT-2 mice have heavier body weight than FAT-1 and FAT-1+2 mice. This could be due to altered food intake and/or energy expenditure. Should be noted in text and discussion either way.

We agree that FAT-2 mice have heavier body weight than FAT-1 and FAT-1+2 mice and this could be due to altered food intake and/or energy expenditure. The food intake was measured during this study. We did not find significant differences in the food intake (**Supplementary Fig. 2a** and **2c**). In the revised manuscript, we included both food intake data (Supplementary Fig. 2a and 2c). and energy metabolism parameters measured using metabolic cages.

The results of energy metabolism experiments have been mentioned in the results section of the revised manuscript as follows:

“The body weight of adult mice at the age of 8 months differed significantly between the genotypes, with the highest body weights for the FAT-2 group and the lowest body weights for the FAT-1 and FAT-1+2 groups (**Fig. 1a**) although no significant differences in the food intake (**Fig. 2a**). Usually, the difference in fat mass observed during the young age would diminish when they become older. We thus followed up the phenotype of these mice for a prolonged WD exposure (16 months). Even after 16 months, FAT-2 group maintained the highest body weight and FAT-1+2 maintained the lowest body weight (**Supplementary Fig. 2b**). Increased body weight was correlated with increases in body fat mass, especially abdominal fat (**Fig. 1b-c**) in the 16-month-old mice. To assess the cause of this persisting body weight difference between groups, especially between FAT-2 and FAT-1+2, we investigated energy metabolism at the age of 16 months. At that point, there were no differences in the food intake between groups (**Supplementary Fig. 2c**). However, FAT-2 mice spent significantly less energy than FAT-1+2 and FAT-1 mice in the dark and light phases (**Supplementary Fig. 2d-e**), together with lower CO₂ production (**Supplementary Fig. 2f-g**), O₂ consumption (**Supplementary Fig. 2h-i**). The respiratory exchange ratio (RER) was unaltered (**Supplementary Fig. 2j-k**). Moreover, it is

notable that this difference in the energy expenditure was not due to a difference in physical activity (**Supplementary Fig. 2l-m**)”.

We have also discussed the energy metabolism related details in the discussion section of the revised manuscript as follows:

“A recent study has shown that male FAT-1 mice fed high-fat diet for 6 weeks at young age showed lean phenotype associated with higher energy expenditure than WT counterparts³⁰. Conversely, the sustained increase in fat mass and eventually body weight observed in FAT-2 mice in this study was likely due to another mechanism, which is the lower energy expenditure (EE). The altered energy homeostasis (lower EE, CO₂ production and O₂ consumption) in the FAT-2 mice with elevated tissue n-6/n-3 ratio could partially be due to impaired browning process³¹ and altered intestinal endocannabinoid system (ECS)³¹. However, this assumption warrants further investigation”.

Figures

13. *Authors must provide legends for the Supplementary Figures.*

We have provided the legends for all the supplementary Figures in the revised manuscript.

14. *Fig 1. Panels A (insert), C, D, H (insert), and I. Font is too small, illegible even at high magnification. What are the units for the scales shown in Panels C and I?*

Thank you. We increased the font size for panels A, C, D, H and I, and removed the insert in the panel H and mentioned the PERMANOVA results in the results section. Mean values of relative abundance were normalized to ‘percentages’ and expressed as heatmaps (panel C and I) so the units for panel C and I were mean (%). We have mentioned these details in the Fig. 2 (Fig. 1 in the original version) legend of the revised manuscript.

15. *Fig 1 & 2. Fig. titles should be changed for greater accuracy. The ratio does not affect the changes, it is the tissue-generated alteration in n-6 and n-3 PUFA production (and thus ratio) that affects the changes. Suggest, “Alterations in tissue-generated n-6/n-3 ratio impact...”*

We understand and agree that the ratio does not affect the changes, it is the tissue-generated alteration in n-6 and n-3 PUFA production (and thus ratio) that affects the changes. We followed your suggestion and modified the Fig. 1 & 2 titles accordingly. To see these changes, please refer Figures 2 & 3 titles in the revised manuscript as we had to change the orders of main Figures as per the reviewer #1. In addition, the individual Figures in the panel c of Fig. 2 (original version) have been considered as individual panels, so each panel has been labelled with a single letter in the revised manuscript (**Fig. 3c-n**).

16. Fig 3 Panel q. The photos need labels. No way to know what mouse groups these come from.

Thank you. We labeled the photos (FAT-2 group) as you advised. Please kindly note that the Fig. 3 in the original submission has been moved as Fig. 1 in the revised manuscript as suggested by reviewer #1. Also, the order of panels in the Fig. 3 has been modified to present first the metabolic disorders and then the markers of metabolic endotoxemia and chronic low-grade inflammation.

17. Fig 1. Panel K. Font is too small, illegible even at high magnification. Order of the groups should match the order in the other panels, especially when the font is nearly impossible to read. Don't know what this data is showing as there is no legend and it is not described in the text.

Thank you very much for bringing this to our attention. Fig. 1k in the original submission is a schematic diagram showing proportions of total n-6 and n-3 PUFA (proportionally divided areas of the bars) and comparing differences in total PUFA (height of the bars) and n-6/n-3 ratio (ratios between given proportions of n-6 and n-3 PUFA) between four genotypes (WT, FAT-1, FAT-2 and FAT-1+2 respectively) fed an identical diet. The main objective of this diagram was to show that four different tissue PUFA profiles were created by using transgenic mice fed an identical diet. Because the total PUFA was the same between WT and FAT-1 and between FAT-2 and FAT-1+2, they were ordered as in the Fig. to make it easy for readers to understand the total PUFA comparisons. To make it legible, we have made a new supplementary Fig. 1 (dot plot showing individual values) with increased the font size. The details of this diagram were described in the supplementary Fig. 1 legend of the revised manuscript.

18. Fig 2. Panels A, C, and I. Font is too small, illegible even at high magnification.

Fig 2a, 2c and 2i of original submission have been modified with increased font size to make them legible. Please refer Fig. 4a and 4c and 4i in the revised manuscript. In addition, Fig. 2d-i of original submission have been modified (dot plot showing individual values) with increased font size. Please refer Fig. 4d-i in the revised manuscript.

Supplementary Data (no page numbers provided)

Page numbers for supplementary data have been provided in the revised manuscript.

19. Animal Diets section: Typographical error(?) - “fat-12” or “Fat12” mice. This nomenclature is inconsistent with the body of the manuscript (where the strain is referred to as FAT-1+2) and is misleading. Looks like “twelve”. Occurs here and other places in the supplementary data document.

Thank you. In the supplementary file, we have rewritten the nomenclature from “fat-12” to “FAT-1+2” to be consistent with the body of the main manuscript.

20. *The complete fatty acid composition details of the diet should be provided, including fatty acid specifics (e.g., amount of linoleic acid, alpha linolenic acid, arachidonic acid, palmitate, etc.) as well as SFA, PUFA and MUFA proportions.*

Please refer “supplementary table 5” in the revised manuscript for the complete fatty acid composition details of the Western diet, including fatty acid specifics as well as SFA, PUFA and MUFA proportions.

21. *Coefficients of variance for the measurement of cytokines and other circulating factors as well as LLOD should be reported for each assay. A description of the assay controls used should also be described.*

Thank you. We have provided the C.V. and LLOD in the methods section for LPS, cytokines, sCD14 and LBP. For all the assays, we had only 5-6 standards including blank (negative control). Please refer the supplementary methods section of the revised manuscript for these details.

22. *Methods here describe insulin tolerance tests (ITT), insulin measurements, and HOMA-IR calculations. None of the data is presented. This data must be presented if the studies were conducted.*

We sincerely apologize that we accidentally included the details of insulin tolerance tests (ITT), insulin measurements, HOMA-IR calculations, serum lipid profiling, liver function tests, hepatic triglyceride measurements and bacterial culture for fecal microbiota analysis in the supplementary methods section although we did not conduct these procedures at all in this study. In fact, in the original version of this manuscript (page 9, lines 3-5), we mentioned that *we examined whether the four genotypes differed in the development of metabolic disorders, including metabolic endotoxemia, systemic inflammation, obesity, fatty liver, glucose intolerance, and cancer.* Because we have already sacrificed these transgenic mice used in this study and there were no more serum samples stored, we have removed these method details in the revised manuscript.

REVIEWERS' COMMENTS:

Reviewer #1 (Remarks to the Author):

The authors' responses to the various comments are relevant, well-argued, and most of the recommendations have been taken into account. Therefore, I consider that this work can be published in this form.

Reviewer #2 (Remarks to the Author):

Summary. The authors have been very responsive to my comments and the manuscript is much improved. Nonetheless, there are a few minor edits that require attention.

Specific comments.

1. Need to define what "balanced tissue ratio of n-6/n-3 PUFA" means in abstract and introduction. The authors have failed to make any changes to the manuscript regarding this point, even after both reviewers questioned it and the authors including much detail about it in their rebuttal. The readership needs to see the explanation, too. Define what you mean by "n-6/n-3 balance" and "balanced n-6/n-3 ratio" etc. in the manuscript abstract and introduction.

2. Response to my original comments 3 & 4 is unacceptable. As n-6 and n-3 PUFAs are essential, by definition, they must be provided in the diet, some might say "supplemented" in the form of types of oils/fat chosen to make the chow. Stating that they diet/chow not "supplemented" is confusing for the typical reader. Please change this sentence:

"They are all fed an identical diet with no need of dietary supplementation with corresponding PUFA."
to

"They were all fed an identical diet, which contained the amount of n-6 and n-3 PUFAs required for mice that do not carry the fat-1 and/or fat-2 transgenes.

3. Page 9, line 214. Authors response in rebuttal letter is still unclear. Suggest the following revision, "Next, from the list of differentially expressed KEGG pathways (FDR corrected $P < 0.05$) represented by the inferred genomic content, we selected those associated with metabolic syndrome (MS), 215 inflammation, bacterial translocation, non-alcoholic fatty liver disease (NAFLD), and cancer 216 (Supplementary Table S1) and then analysed the abundance of these pathways among the different mouse genotypes (Supplemental Methods)." Many methods in supplemental are not mentioned in main text. Should be referenced there.

4. I understand from the authors that they do not want to respond to my comment. "The multi-/inter-omics results sections are overly lengthy and should be abbreviated significantly. The detail is too great for this journal and readership, in my opinion." I defer to the editor.

5. Supplemental Table 1. Title is grammatically incorrect. Change "its" to "their".

RESPONSE TO REVIEWERS' COMMENTS:

Reviewer #1 (Remarks to the Author):

The authors' responses to the various comments are relevant, well-argued, and most of the recommendations have been taken into account. Therefore, I consider that this work can be published in this form.

In the revised manuscript, we defined what we mean by “n-6/n-3 balance” and “balanced n-6/n-3 ratio” in the appropriate place of abstract and introduction sections of the manuscript.

Reviewer #2 (Remarks to the Author):

Summary. The authors have been very responsive to my comments and the manuscript is much improved. Nonetheless, there are a few minor edits that require attention.

Specific comments.

1. Need to define what “balanced tissue ratio of n-6/n-3 PUFA” means in abstract and introduction. The authors have failed to make any changes to the manuscript regarding this point, even after both reviewers questioned it and the authors including much detail about it in their rebuttal. The readership needs to see the explanation, too. Define what you mean by “n-6/n-3 balance” and “balanced n-6/n-3 ratio” etc. in the manuscript abstract and introduction.

In the revised manuscript, we defined what we mean by “n-6/n-3 balance” and “balanced n-6/n-3 ratio” in the appropriate place of abstract and introduction sections of the manuscript

2. Response to my original comments 3 & 4 is unacceptable. As n-6 and n-3 PUFAs are essential, by definition, they must be provided in the diet, some might say “supplemented” in the form of types of oils/fat chosen to make the chow. Stating that they diet/chow not “supplemented” is confusing for the typical reader. Please change this sentence:

“They are all fed an identical diet with no need of dietary supplementation with corresponding PUFA.”

to “They are were all fed an identical diet, which contained the amount of n-6 and n-3 PUFAs required for mice that do not carry the fat-1 and/or fat-2 transgenes.

The sentence suggested by the reviewer is not what we want to say. What we really try to express is that all animals used in our study were fed an identical diet, to which we did not need to add (supplement) extra amount of PUFA (n-6 or n-3 fatty acids) in order to create high levels of PUFA in animal tissues. For example, we did not need to supply an additional amount of n-3 PUFA in the diet for fat-1 mice to have higher levels of n-3 PUFA in their tissues. Similarly, we did not need to add an extra amount of n-6 PUFA in the diet for fat-2 mice to have higher levels

of n-6 PUFA in their tissues, as these transgenic mice can make specific PUFA themselves. (Usually, to create different PUFA profiles in animals, we need to feed animals with special diets supplemented with specific PUFA). I hope we have clarified enough now (sorry for the confusion). The common diet we used for all animals contains basic levels of PUFA should not be regarded as a supplemented diet. So, we think that we can still use our original sentence. To avoid the confusion, we added the sentence you suggested in the 'animal diets' section (please refer methods in the revised manuscript).

3. Page 9, line 214. Authors response in rebuttal letter is still unclear. Suggest the following revision, “Next, from the list of differentially expressed KEGG pathways (FDR corrected $P < 0.05$) represented by the inferred genomic content, we selected those associated with metabolic syndrome (MS), 215 inflammation, bacterial translocation, non-alcoholic fatty liver disease (NAFLD), and cancer 216 (Supplementary Table S1) and then analysed the abundance of these pathways among the different mouse genotypes (Supplemental Methods).” Many methods in supplemental are not mentioned in main text. Should be referenced there.

Thank you very much. We followed your suggestion and revised the response accordingly in the revised manuscript. In addition, we moved almost all the methods from the supplementary file to the "methods" section of the main manuscript.

4. I understand from the authors that they do not want to respond to my comment. “The multi-/inter-omics results sections are overly lengthy and should be abbreviated significantly. The detail is too great for this journal and readership, in my opinion.” I defer to the editor.

Because the reviewer still feels that the detail is too great for this journal and readership, we significantly abbreviated the "the multi- /inter-omics results sections in the revised manuscript.

5. Supplemental Table 1. Title is grammatically incorrect. Change “its” to “their”.

We changed “its” to “their” in the Supplementary Table 1 of the revised manuscript